# Spectral hallmark of auditory-tactile interactions in the mouse somatosensory cortex

Manning Zhang[1,3], Sung Eun Kwon[2,4], Manu Ben-Johny[1,5], Daniel H. O'Connor [2] & John B. Issa [1,6]*

To synthesize a coherent representation of the external world, the brain must integrate inputs across different types of stimuli. Yet the mechanistic basis of this computation at the level of neuronal populations remains obscure. Here, we investigate tactile-auditory integration using two-photon $Ca^{2+}$ imaging in the mouse primary (S1) and secondary (S2) somatosensory cortices. Pairing sound with whisker stimulation modulates tactile responses in both S1 and S2, with the most prominent modulation being robust inhibition in S2. The degree of inhibition depends on tactile stimulation frequency, with lower frequency responses the most severely attenuated. Alongside these neurons, we identify sound-selective neurons in S2 whose responses are inhibited by high tactile frequencies. These results are consistent with a hypothesized local mutually-inhibitory S2 circuit that spectrally selects tactile versus auditory inputs. Our findings enrich mechanistic understanding of multisensory integration and suggest a key role for S2 in combining auditory and tactile information.

[1] Department of Biomedical Engineering, The Johns Hopkins University School of Medicine, Baltimore, MD 21205, USA. [2] Solomon H. Snyder Department of Neuroscience, The Johns Hopkins University School of Medicine, Kavli Neuroscience Discovery Institute, and Brain Science Institute, Baltimore, MD 21205, USA. [3] Present address: Department of Biomedical Engineering, Washington University in St. Louis, St. Louis, MO 63130, USA. [4] Present address: Department of Molecular, Cellular and Developmental Biology, University of Michigan, Ann Arbor, MI 48109, USA. [5] Present address: Department of Physiology and Cellular Biophysics, Columbia University, New York, NY 10032, USA. [6] Present address: Department of Neurobiology, Northwestern University, Evanston, IL 60201, USA. *email: john.issa@gmail.com

Our understanding of the external world is derived from what our senses can tell us[1]. We can enrich this representation of the environment by combining inputs across disparate sensory modalities[2]. However, maintaining a coherent percept requires active moment-to-moment integration of multisensory inputs along with resolution of inconsistencies between modalities[3]. How this active process is carried out at the level of neural populations is an open question.

Two stimulus modalities that are deeply entangled are sound and touch. Mechanical forces generated by physical contact not only activate somatosensory mechanoreceptors but also generate acoustic waves that are detected by the machinery of the auditory system[4]. Thus these stimuli are often highly correlated in the environment[5]. A significant behavioral consequence of this interaction is observed when tactile and auditory inputs are not in register: as a human subject touches a surface, if a sound is played that is different from what is expected from the tactile sensation of the surface, the reported roughness of the surface is altered in a manner dependent on the sound frequency[6]. This "parchment-skin illusion" points to the deep bond shared by these modalities and hints at the centrality of stimulus frequency as a parameter that may bind them together[7,8]. Consistent with this idea, the secondary somatosensory cortex of mice can respond to sound[9]. Furthermore, retrograde tracing in Mongolian gerbils has revealed direct synaptic projections from primary somatosensory cortex to primary auditory cortex[10] and single unit recordings in behaving monkeys have shown that neurons in anterior parietal areas respond to task-related auditory cues[11], hinting at a cohesive sound-touch integration circuit within cortex.

In this study, we leverage these observations to probe the neural correlates of multisensory integration in the neocortex. We aim to determine how stimulus frequency is encoded by neurons in the somatosensory system, whether concurrent sound can influence these responses, and how this information is carried across a population of neurons. Thus, we performed large-scale two-photon $Ca^{2+}$ imaging in multiple cortical regions: the primary (S1) and secondary (S2) somatosensory cortices, the insular somatosensory field (ISF), and the auditory cortex. We find that neurons in S1 and S2 encode the frequency at which a whisker is deflected. Concurrent sound modulates these responses, with responses to low tactile frequencies often completely abolished. In S2, we find a small number of sound-selective neurons whose responses are reciprocally attenuated by high-frequency whisker deflections. These frequency-dependent interactions point towards a spectrally-dependent mutually inhibitory circuit between touch-selective and sound-selective neurons and shed light on the neural circuits that may underlie the computations involved in multisensory integration.

## Results

### Widefield $Ca^{2+}$ imaging localizes tactile and auditory areas.
We performed widefield $Ca^{2+}$ imaging of the left temporal lobe of mouse neocortex to identify regions that respond to either tactile or auditory stimuli individually. Transgenic mice expressing the $Ca^{2+}$-indicator GCaMP6s under pan-neuronal promoters[12–14] were implanted with a chronic cranial window exposing up to 5 mm of the left hemisphere[15,16]. This approach allowed us to capture the majority of the auditory cortex along with the primary somatosensory (S1), the secondary somatosensory (S2), and the insular somatosensory (ISF; insular somatosensory field) cortices all in the same window. Mice were habituated to head fixation while running on a treadmill. During widefield fluorescence imaging, either tactile or auditory stimuli were delivered to the contralateral target whisker or ear, respectively (Fig. 1a). In response to a sinusoidal 128 Hz stimulus applied to the C2

whisker, transient increases in GCaMP6s fluorescence were observed in somatosensory areas, with smaller responses in the auditory fields (Fig. 1b, c). A spatial map of these changes, which averages the change in fluorescence over 1 s of the response for stimulus frequencies ranging from 36 to 128 Hz (Fig. 1d), highlights three distinct locations that were strongly responsive to tactile stimuli. Based on stereotaxic coordinates (predicted location of C2 barrel of S1 indicated by red cross), the most medial locus was identified as S1, the middle locus corresponded to S2, and the most lateral locus was consistent with the location of ISF identified previously with intrinsic optical imaging[17]. In response to auditory stimuli composed of sinusoidal amplitude modulated (SAM; Supplementary Fig. 1) tones, we observed robust fluorescent responses in regions corresponding to the auditory cortex (Fig. 1e, f). SAM tones were chosen to approximate a correspondence between the envelope of auditory stimuli and the frequency of the sinusoidal waveform of the tactile stimuli used throughout this study (in the range of 2 to 128 Hz). Interestingly, we observed a small transient in ISF, consistent with overlap of the nearby insular auditory field (Fig. 1g, h)[17,18].

### Tuning to the frequency of tactile stimuli in S1 and S2.
Having located each cortical area within our cranial window, we next examined the responses of individual layer 2/3 neurons across multiple somatosensory regions. Using two-photon $Ca^{2+}$ imaging, we monitored the activity of hundreds of neurons within a field-of-view (FOV) spanning $465 \times 465 \, \mu m^2$, a region sufficiently large to include the C2 barrel column in S1 (Fig. 2a)[19]. To measure evoked tactile responses, we chose sinusoidal piezo-driven tactile stimuli with frequencies ranging from 2 to 128 Hz presented in random order. The upper limit of 128 Hz was imposed to avoid any piezo-generated sound within the hearing range of mice (see Methods and Supplementary Fig. 1). We found 16% of neurons in S1 were responsive to passive whisker stimulation (598 of 3723 neurons, 16 imaging fields from 8 mice). For comparison, a previous study found 16.8% of layer 2/3 excitatory neurons in the principal column responded to touch while 39[19] or 37%[16] showed task-related tactile responses. Figure 2b displays individual baseline-normalized fluorescent responses $\left( \delta \hat{F} = \Delta F/F_0 \right)$ of two exemplar neurons to tactile stimuli of increasing frequency (multiple trials in gray; ensemble average in black, sorted according to stimulus frequency). A normalized tactile response for each stimulus was then estimated as the ratio of the mean fluorescence change $\left( \delta \hat{F}_{mean}(i) \right)$ for the given stimulus $(i)$ to the maximal fluorescence change $(\max_i \delta \hat{F}_{mean}(i))$ observed across all stimuli. Plotting the normalized tactile response as a function of frequency revealed a normalized tactile tuning curve (Fig. 2c). The best tactile frequency (BTF, see Methods and Supplementary Fig. 2) served as a convenient measure of the tuning preference of individual neurons. Figure 2d shows the overall distribution of BTF for tactile-responsive neurons in S1. Neurons with low BTFs tended to have biphasic tuning curves while neurons with high BTFs exhibited monotonically increasing responses (Fig. 2c), although we could not exclude the possibility that high-BTF neurons may have responses that decrease at frequencies beyond 128 Hz. In S1, neurons with high BTFs (>60 Hz) outnumbered low-BTF neurons 6-fold in relative prevalence.

We further assessed tactile responses of neurons within S2 (Fig. 2e), ~1 mm lateral to S1 based on widefield $Ca^{2+}$ imaging results (Fig. 1d). We found that 9% of all neurons were responsive to tactile stimuli (1313 of 14051 neurons, 67 imaging fields from 19 mice). Again, examination of tactile tuning curves revealed the presence of neurons with varying response profiles (Fig. 2e, f). Neurons with low BTFs were modestly but significantly more

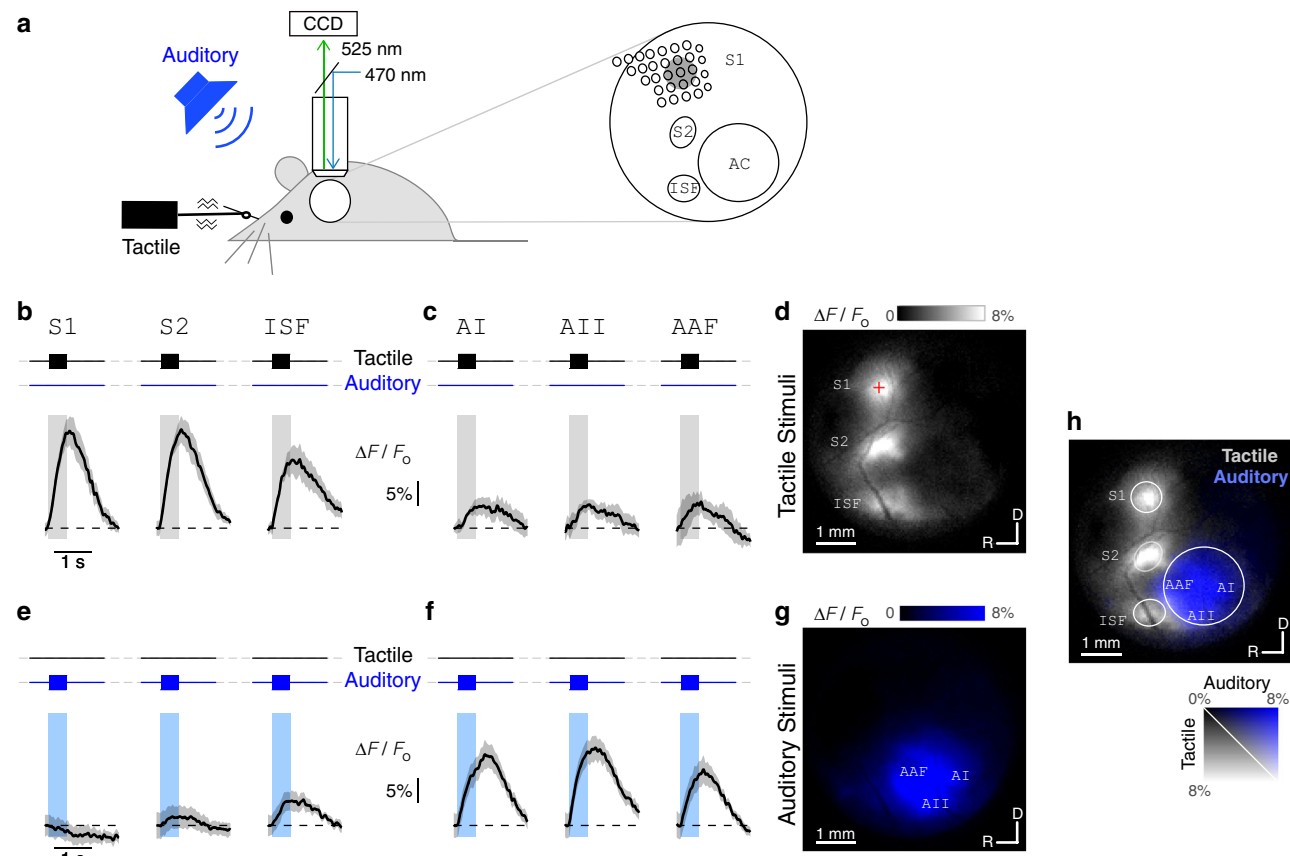

**Fig. 1 Responses to tactile and auditory stimuli under widefield Ca$^{2+}$ imaging. a** Widefield imaging set-up. Tactile and auditory stimuli are delivered to a head-fixed awake mouse. Piezo coupled to right C2 whisker provides vibration in rostral-caudal direction while speaker delivers sinusoidal amplitude-modulated (SAM) tones to the right ear. 5 mm chronic cranial window over the left temporal cortex (white circle) spans somatosensory and auditory areas. GCaMP6s fluorescence (470 nm illumination and 525 nm emission) collected by CCD camera through 4× objective. S1: primary somatosensory cortex; S2: secondary somatosensory cortex; ISF: insular somatosensory field; AC: auditory cortex. **b** Responses to tactile stimuli in somatosensory areas. Average of four single trial Ca$^{2+}$ transients evoked in S1, S2, and ISF in response to 128 Hz whisker stimulation. Black trace shows fluorescence activity ($\Delta F/F_0$) averaged over 200 × 200 µm$^2$ regions in each of these areas. Standard error shown as shaded region. Vertical gray bar represents 500 ms stimulus period. **c** Responses to tactile stimuli in auditory areas. Same format as **b**. **d** Ca$^{2+}$ activity during presentation of tactile stimuli reveals location of somatosensory areas. Spatial map depicts average response to 36–128 Hz tactile stimuli (7 individual frequencies), taken as average of $\Delta F/F_0$ signal over first 1 s after stimulus onset. Responsive regions correspond to S1, S2, and ISF, as labeled. Red cross denotes 1.3 mm posterior and 3.5 mm lateral of bregma. Dorsal-rostral orientation and 1 mm scale bar shown at bottom. **e** Responses to 9.5 kHz SAM tone in S1, S2 and ISF averaged over six trials. Vertical blue bar represents 500 ms stimulus period. **f** Responses to auditory stimuli in auditory areas. Same format as **e**. **g** Ca$^{2+}$ activity during presentation of auditory stimuli reveals location of auditory areas. Image shows average response of 3–48 kHz SAM tone stimuli (7 individual frequencies). Responsive regions correspond to AI (primary auditory cortex), AII (secondary auditory cortex), and AAF (anterior auditory field), as labeled. **h** Overlay of responses to tactile (gray, as shown in **d**) and auditory (blue, as shown in **g**) stimuli highlighting relative location of somatosensory and auditory fields in the same window. Rostral-caudal orientation and $\Delta F/F_0$ scale bar shown at bottom right.

prevalent in S2 compared to S1 (19.8% for S2 and 15.9% for S1, $p$ < 0.05 by Chi-square proportion test; Fig. 2g). Neurons with high *BTFs* nonetheless outnumbered low-*BTF* neurons by ~4-fold. The coarse similarity in response characteristics of both S1 and S2 neurons indicates that tactile frequency tuning may be a general feature for encoding tactile sensation in multiple regions of the somatosensory cortex.

**Concurrent auditory input modulates tactile responses in S2.** To identify possible multimodal integration, we explored whether concurrent auditory stimuli could modify tactile responses in the somatosensory cortex, focusing initially on S2 given its putative role in higher-order tactile processing[20–22]. Accordingly, we measured fluorescence changes of single neuron responses in S2 under three stimulus conditions: (1) tactile stimuli alone, (2) auditory stimuli (SAM tones) alone, and (3) tactile and auditory

stimuli presented concurrently. Figure 3 displays responses of three tactile-selective neurons within a single FOV demonstrating various multisensory effects. Neuron 1 exhibited robust responses to high tactile frequencies (black, Fig. 3b) that were enhanced upon concurrent presentation of acoustic stimuli (red, Fig. 3b). The normalized responses obtained with tactile stimuli alone and with concurrent tactile and auditory stimuli are overlaid in Fig. 3c. By contrast, neuron 2, which was also tuned to the high-frequency tactile stimuli, demonstrated inhibition of this response when concurrent auditory stimulation was added (Fig. 3b, c). Finally, neuron 3 exhibited robust responses when probed with tactile stimuli alone (black, Fig. 3b) but was largely silenced when tactile and auditory stimuli were presented concurrently (red, Fig. 3b). All three neurons were nonresponsive to all sound stimuli tested when sound stimuli were presented alone (blue, Fig. 3b). These results illustrate a powerful mode of multisensory processing whereby the response of individual cortical neurons to

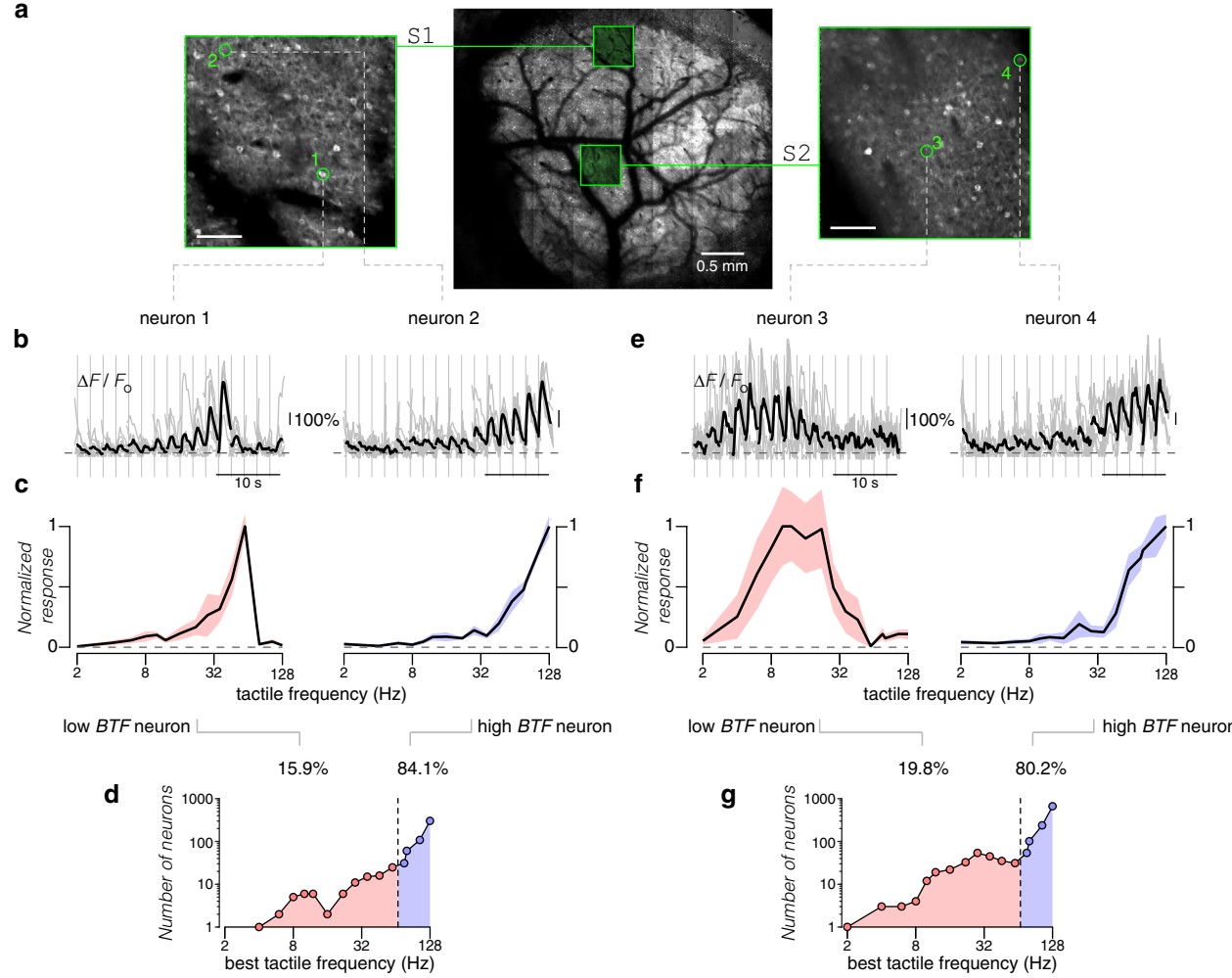

**Fig. 2 Responses to tactile stimuli in somatosensory cortices under two-photon Ca²⁺ imaging. a** Map highlights location of FOVs centered on C2 barrel within S1 and S2 among the entire cranial window (formed by tiling 7 × 8 individual FOVs). Scale bar on individual FOVs: 100 μm. **b** Ca²⁺ activity ($\Delta F/F_0$) of two exemplar S1 neurons during presentation of tactile stimuli at different frequencies. Stimuli were played in random order and responses sorted by increasing frequency (2–128 Hz). Gray traces show $\Delta F/F_0$ for individual trials. Black traces show average $\Delta F/F_0$ across all trials (7 repeats). Vertical gray line denotes start of each stimulus. **c** Frequency tuning curves of individual neurons shown in **b**. Responses are averages over a 600 ms period of the deconvolved $\Delta F/F_0$ traces, which are then normalized to the maximum response across all stimuli for each neuron. Standard error shown as shaded region. Frequency of tactile stimuli shown on the x-axis. **d** Population distribution of *BTF* (see Methods) for S1 neurons. Black circles show number of neurons tuned to each frequency tested. To facilitate comparison with S2, neurons with best frequencies no higher than 60 Hz were categorized as low *BTF* neurons (pink) while those with best frequencies above 60 Hz were categorized as high *BTF* neurons (blue). Both x and y axes plotted on a logarithmic scale. **e** Ca²⁺ activity induced by tactile stimuli in S2 for two exemplar neurons. Average across 6 trials. **f** Frequency tuning curves for the exemplar neurons shown in **e**. **g** Population distribution of *BTF* in S2.

their primary sensory modality is modulated by a concurrent secondary sensory input. These effects are diverse, allowing for neuron-specific computations of multisensory interactions.

To assess the prevalence of each mode of tactile-auditory integration in S2, we combined results from 67 FOVs across 19 animals. For all tactile-selective neurons in S2, we systematically probed for statistically significant changes in tactile responses by concurrent auditory stimulation (2-way ANOVA with a pre-established criterion of $p < 0.05$ for significance). We found that most neurons had tactile responses that were either inhibited or unchanged by sound, with neurons tuned to lower frequencies more likely to be inhibited (Fig. 3d). To probe the overall potency of sound modulation, we quantified the net change in aggregate tactile responses (average response elicited across all tactile frequencies) when evoked by concurrent tactile and auditory stimuli ($\Delta R$, see Methods) (Fig. 3e). The relative strength of sound-driven inhibition was most potent in neurons tuned to

lower frequencies (*BTF* up to 60 Hz: $\Delta R = 43\%$) in comparison to neurons tuned to higher frequencies (*BTF* > 60 Hz: $\Delta R = 31\%$; $p < 0.001$ by Wilcoxon rank-sum test) (Fig. 3e). In this manner, sound-driven inhibition appeared to be both more prevalent and more prominent in neurons tuned to lower frequencies, indicating a possible tactile frequency dependence in multisensory interactions.

**Dissociation of motor behavior from sound-driven modulation.** Ongoing locomotor activity can drastically alter activity in sensory cortices[23,24]. To determine whether the observed sound-driven inhibition in S2 may be associated with differences in motor activity and not directly caused by sound stimulation (Fig. 4a), we recorded the animal's movement velocity on a treadmill during two-photon imaging of neural activity and used a generalized linear model (GLM) to compare the influence of

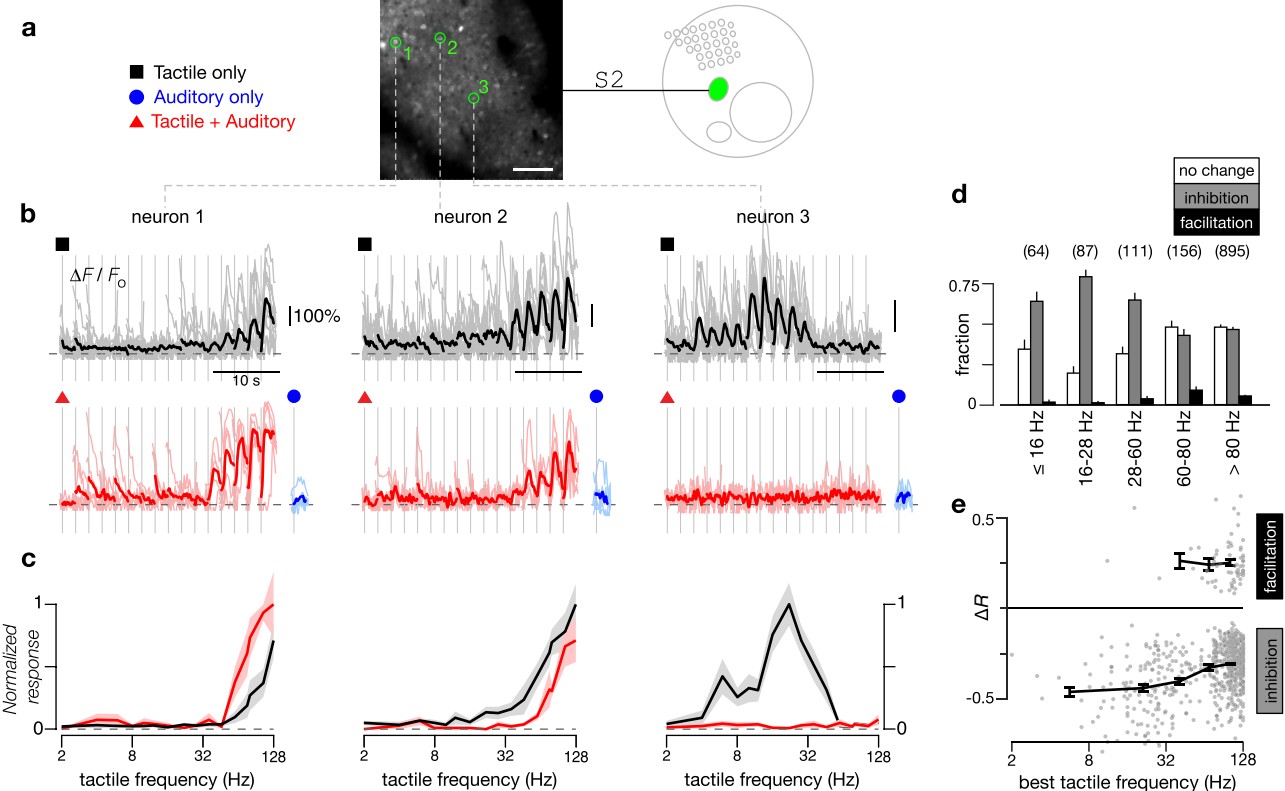

**Fig. 3 Sound-modulated response to tactile stimuli in S2. a** Baseline fluorescence image of two-photon imaging field located in S2, as illustrated in schematic map. Symbols denote different stimuli tested. Exemplar neurons to be scrutinized in panels **b**, **c** are highlighted by green circles (panel). Scale bar: 100 μm. **b** Responses of exemplar neurons to tactile stimuli alone (black traces, 11 repeats) or to combined tactile and auditory stimuli (red traces, 5 repeats). In both cases, tactile frequency ranged from 2 to 128 Hz. For the combined stimulus, a concurrent auditory stimulus (10 kHz tones with 64 Hz SAM envelope at 20 dB attenuation) was added to the tactile stimuli. In addition, blue traces show averaged responses to the same auditory stimulus alone. For all cases, individual trials shown as light traces and response averages shown as dark traces. **c** Frequency tuning curves of the normalized response to tactile stimuli and combined tactile plus auditory stimuli for exemplar neurons shown in **b**. Black line shows the normalized tuning curve for responses to tactile stimuli. Red line shows the normalized tuning curve for responses to tactile plus auditory stimuli. Standard error shown as shaded gray and pink regions. **d** Fraction of neurons in S2 exhibiting either no change, a decrease (inhibition), or an increase (facilitation) in responses when SAM tones were added to the tactile stimuli, with error bars indicating standard error for a multinomial distribution. Criteria determined by 2-way ANOVA ($p < 0.05$). Neurons binned into one of five categories based on their *BTF*; number of neurons in each category indicated by numbers above the bar plots. Results pooled across 67 FOVs from 19 mice. **e** Percentage change in response of S2 neurons to tactile stimuli ($\Delta R$) when SAM tones were added as a function of *BTF*, shown for inhibited or facilitated neurons as described in **d**. Thick line and error bars show mean and standard error of $\Delta R$ among inhibited or facilitated neurons within each category from **d**, and are only shown if at least 3 neurons were found within that category.

sound stimulation with motor activity on tactile responses. Tactile responses in exemplar neurons were divided into three categories of movement velocity, two categories of sound stimulation (off or on), and four categories of tactile stimulation frequency (Fig. 4b, c). For neuron 1 (Fig. 4b), responses are not appreciably altered by locomotion (compare rows), indicating that sound-driven inhibition of tactile responses cannot be explained by ongoing behavior. For neuron 2 (Fig. 4c), running diminished responses; however, sound also inhibited this neuron's tactile responses. Using a GLM to quantify the contribution of sound (coefficient *c*, z-scored) and locomotion (coefficient *d*, z-scored) to each neuron's tactile responses (see Methods), we found that neuron 1 was at best weakly inhibited by running velocity ($d = -0.3$) while neuron 2 was strongly inhibited ($d = -2.9$) (Fig. 4d). Importantly, in both cases sound is a strong inhibitory factor ($c = -3.4$ and $-3.5$, respectively).

Across the population of tactile-responsive neurons in S2, those previously identified as unperturbed by sound ('no change') were found to have sound amplitudes near zero in the GLM ('inhibited'), while those identified as inhibited by sound tended to have large negative values for the sound coefficient (Fig. 4e, f).

Interestingly, for sound-inhibited tactile-responsive neurons, neurons tuned to lower tactile frequencies were inhibited by concurrent running behavior while neurons tuned to higher tactile frequencies were facilitated by running (blue curve, Fig. 4g). Across all *BTF*s, however, the coefficient for sound modulation was consistently of a larger amplitude (red curve, Fig. 4g). While running velocity tended to increase in over half the FOVs in S2 (58 of 108 FOVs, two-sample *t*-test comparing running during stimulus presentations +/− sound, criteria: $p < 0.05$), the coefficient for sound modulation was uncorrelated with the change in running behavior (Fig. 4h, $p = 0.85$ for Pearson's correlation). In summary, consideration of locomotive behavior does not explain away the contribution of sound to the modulation of tactile responses (Fig. 4i).

Active whisking could also influence the responses of somatosensory neurons to passive whisker stimulation. To test the possibility that the observed sound-driven inhibition can be attributed to an indirect pathway involving active whisking, we transected the buccal branch of the facial nerve in a subset ($n = 2$) of mice, thus severely diminishing the ability of the mouse to whisk[25]. We imaged whisker-responsive S2 neurons in these mice

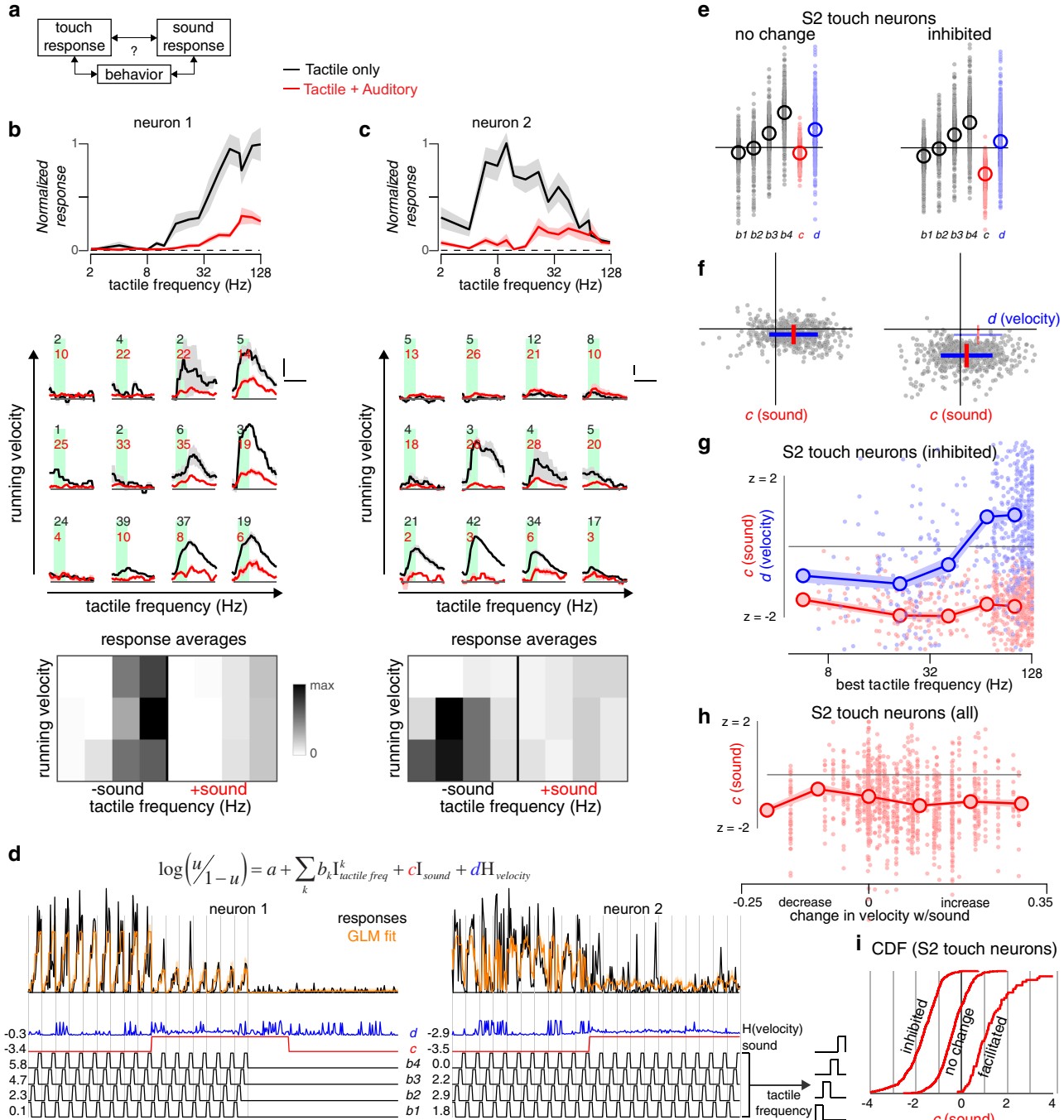

**Fig. 4 Sound-driven inhibition of tactile responses is not explained by locomotive behavior. a** Behavior could potentially mediate the relationship between sound and touch responses in S2. **b** Tactile responses (black, 9 repeats) are significantly suppressed by concurrent presentation of sound (red, 7 repeats) in an exemplar S2 neuron. Post-stimulus response averages are stratified by tactile frequency (horizontal axis), running behavior (vertical axis), and the absence or presence of sound (black and red, respectively), and the average response for each category is represented as an image underneath. **c** Responses for a second exemplar neuron, in the same format as **b**. Tactile stimuli alone: 10 repeats. Tactile stimuli with concurrent sound: 11 repeats. **d** GLM used to model responses of the two exemplar neurons as a summation of tactile frequency (black, divided into 4 frequency categories), running behavior (blue, passed through a nonlinear function), and the absence or presence of sound (red). Each point is the amplitude of response to a stimulus presentation (black) and the GLM fit is overlaid (orange). **e** GLM coefficients (z-scored) across the population of S2 neurons with no change or inhibition of tactile responses with the addition of sound (see Fig. 3d). Circles represent averages. **f** Sound and running velocity coefficients for each neuron plotted against each other as a scatter plot. Red and blue lines represent mean±SEM (thin lines reproduce lines from other subpanel). **g** Sound and running velocity coefficients for the sound-inhibited S2 population, binned by best tactile frequency. **h** Sound coefficient as a function of the change in running velocity associated with the sound stimulus itself (after passing through logarithm function described in Methods). Running velocity increased in 58 of 108 FOVs in S2 whereas 5 FOVs showed a decrease (assessed by two-sample *t*-test comparing running during stimulus presentations +/− sound, criteria: $p < 0.05$). Mean sound coefficient found was insensitive to the concomitant change in running behavior of the animal ($p = 0.85$, Pearson's correlation). Individual neurons: dots; means: circles; shaded region: ±SEM. **i** Cumulative distribution of the sound coefficient for three populations of S2 tactile-responsive neurons, as identified in Fig. 3d.

before and after cutting the nerve (Supplementary Fig. 3). Dramatic sound-driven inhibition of tactile responses was still observed (Supplementary Fig. 3b, c), and the proportion of neurons exhibiting sound-driven inhibition was preserved (58% before transection and 54% after transection, $p > 0.5$ by Chi-square proportion test; Supplementary Fig. 3d, e). Thus, active whisking is not crucial to the observed multimodal interactions.

**Dependence of integration on modality and sound properties**. In a subset ($n = 3$) of mice, we examined the possibility that these interactions may be generic across sensory modalities by using a number of different auditory stimuli and comparing the resultant changes in tactile responses to those induced by visual stimulation (Fig. 5). Responses are shown for two exemplar neurons. In neuron 1, tactile responses were suppressed by both auditory and visual modalities, while responses in neuron 2 were unchanged by sound but facilitated by visual stimuli (Fig. 5b, c). Across the population of S2 neurons, we found that the effect of sound was similar across the different sounds tested, with similar effects regardless of the specific sound (tone versus noise) or modulation frequency. Roughly 75% of neurons responded consistently. By contrast, light produced quantitatively different effects, with roughly 50% of neurons showing a different pattern of modulation (Fig. 5e). Thus, sound-driven inhibition of tactile responses appears to be modality-specific in some neurons but modality-independent in others. Modulation in this latter group of neurons may be mediated by a general mechanism that is common to visual and auditory stimuli, such as changes in attentional state.

Next, we examined whether sound-driven inhibition of tactile responses may depend on the specific properties of the sound stimulus used. Along with attaining the tactile frequency tuning curves of neurons in the absence and presence of a concurrent sound stimulus (Supplementary Fig. 4a, d), we measured responses to SAM tones in the absence and presence of tactile stimuli while varying either the carrier frequency (Supplementary Fig. 4b, c) or modulation frequency (Supplementary Fig. 4e, f) of the sound. Responses of sound-selective neurons in S2 were strongest for carrier frequencies between 6 and 24 kHz (Supplementary Fig. 4b, c, blue curves), while touch responses in tactile-selective neurons were selectively inhibited by those same carrier frequencies (orange curves). Meanwhile, the population of S2 sound-selective neurons, while individually dependent on modulation frequency (33 of 63 neurons), collectively exhibited only a small increase in their response as modulation frequency increased (Supplementary Fig. 4e, f, blue curves). Touch responses in tactile-selective neurons were individually agnostic to modulation frequency, with only 20 of 259 neurons showing any dependence (Supplementary Fig. 4e, f, orange curves). Yet, as a population, a weak trend was observed, with tactile-selective neurons more strongly inhibited by higher sound-modulation frequencies, which was weakly but significantly anti-correlated to the responses of sound-selective neurons (Supplementary Fig. 4f). These results indicate that the observed sound-driven inhibition of tactile responses in S2 is proportional to the total sound-driven response and is not a peculiarity of the precise frequencies that compose the auditory stimulus.

**Concurrent sound modulates tactile responses in S1 and ISF**. Primary sensory cortices are believed to be predominantly involved in unimodal processing, but the degree of multimodal interactions has long been debated. As a wealth of studies have illustrated potential multisensory interactions in the primary cortices[26–28], we examined whether tactile responses in mouse S1 are modulated by concurrent auditory stimuli. Figure 6b, c shows fluorescence recordings for three exemplar tactile-selective

neurons within a single FOV in response to tactile stimuli alone or concurrent tactile and auditory stimuli. Neuron 1 exhibited minimal change to its tactile tuning relation following addition of concurrent sound. By contrast, tactile responses of both neurons 2 and 3 were inhibited by sound. All three neurons were non-responsive to acoustic stimuli when presented alone (blue traces, Fig. 6b). At the population level, unlike neurons in S2, the tactile responses for a majority of high-$BTF$ (69%) and low-$BTF$ (75%) neurons in S1 were unaltered by auditory stimuli (48 and 28% for S2, respectively; $p < 1e-10$ by Chi-square proportion test for both categories when compared with S1). Nonetheless, we identified a substantial number of tactile neurons exhibiting sound-driven inhibition and a small number with sound-driven facilitation. The overall proportions of neurons exhibiting sound-modulation of tactile responses is summarized in Fig. 6d. Like sound-modulated neurons in S2, the degree of inhibition was stronger for neurons with lower $BTF$s (Fig. 6e) ($p < 0.01$ by Wilcoxon rank-sum test), while the degree of sound-driven facilitation was similar between the two types of tactile neurons ($p > 0.1$). These results clearly show the presence of sound-driven modulation of neurons in the primary somatosensory cortex.

We also measured response properties of neurons in the ISF, a poorly understood cortical region found lateral to S2. Here, we observed robust responses to tactile stimulation (Supplementary Fig. 5). In contrast to S1 and S2, we found a dearth of low-$BTF$ neurons, with 18-fold more high-$BTF$ neurons (Supplementary Fig. 5d). The percentage of tactile neurons suppressed by auditory stimulation resembled that of S2 (Supplementary Fig. 5e). The presence of similar modes of tactile-auditory interactions across S1, S2, and ISF suggests that multimodal integration occurs across a distributed somatosensory cortical network.

**Sound preferentially inhibits low frequency tactile responses**. Given the prevalence of sound-driven inhibition of tactile responses observed for neurons in the somatosensory cortex, we next sought to dissect spectral features of multimodal processing. We scrutinized tactile tuning relations in the presence and absence of concurrent auditory stimuli for four neurons with varying $BTF$ within a single FOV in S2. Neuron 1, tuned to tactile frequencies between 8 to 32 Hz, was inhibited across all tactile frequencies (Fig. 7a, b). Interestingly, for neurons 2 and 3, sound-driven inhibition was nearly complete at low tactile frequencies (below 32 Hz) while the high-frequency tactile responses were largely spared. Similar effects were observed for neuron 4, which was responsive to high tactile frequencies, suggesting that sound-driven modulation is exquisitely dependent on the frequency of tactile stimulation: responses to low-frequency tactile stimuli are preferentially suppressed while responses to high-frequency tactile stimuli are relatively preserved.

To quantify the tactile spectral properties of sound-driven inhibition, we assigned neurons to five groups according to their $BTF$s (Fig. 7c). Normalized tactile tuning curves (Fig. 7c) in the absence (black) and presence (red) of concurrent sound stimuli were averaged for the five categories. To quantify the strength of sound-driven inhibition, we computed the ratio of normalized tactile responses in the presence of sound stimuli to those in the absence of sound stimuli at each individual tactile frequency ($r_{sound}$). Across all groups, $r_{sound}$ increased monotonically with increasing tactile stimulus frequency. Low frequency tactile responses were strongly diminished ($r_{sound} \sim 0.1$) by sound stimuli while high frequency tactile responses were more weakly perturbed ($r_{sound} \sim 0.8$) (Fig. 7d). The tactile frequency for half-inhibition increased only weakly as a function of the $BTF$ for the given class of neurons (Fig. 7g). Overall, these results show that sound-driven inhibition

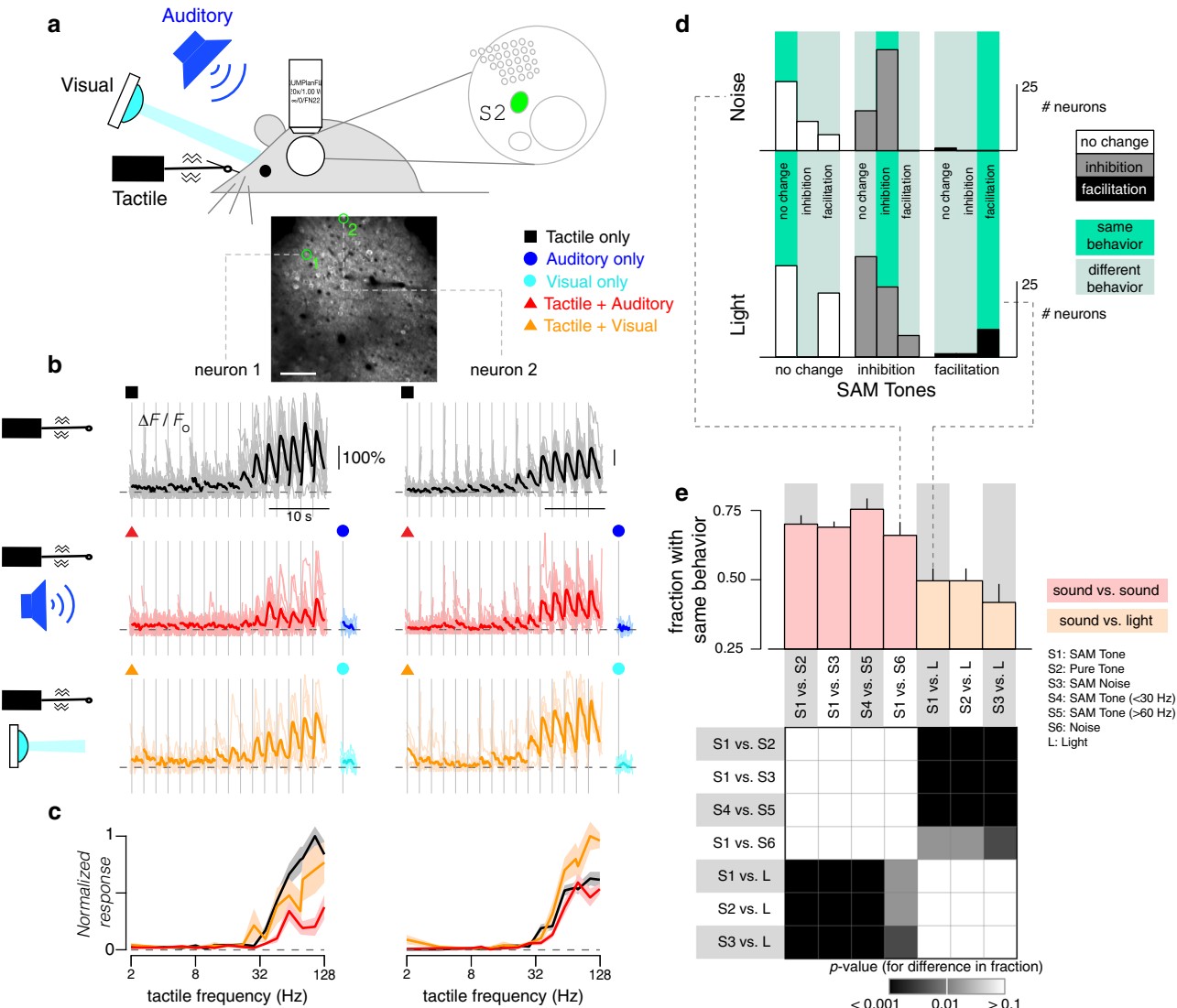

**Fig. 5 Modality-specific multimodal modulation of tactile responses. a** Tactile, auditory, and visual stimuli were presented either individually or concurrently while two-photon imaging was performed in S2. Baseline fluorescence image of imaging field located in S2 with exemplar neurons highlighted in green. Scale bar: 100 μm. **b** Responses of exemplar neurons to tactile stimuli alone (black traces, 16 repeats), combined tactile and auditory stimuli (red traces, 13 repeats), or to combined tactile and visual stimuli (orange traces, 6 repeats). Blue traces (7 repeats) show averaged responses to 10 kHz tones with 64 Hz SAM envelope at 40 dB attenuation and cyan traces show averaged responses to visual stimuli alone (LED pulsed on for 500 ms). Responses from individual trials are shown as light traces. **c** Frequency tuning curves of the normalized response to tactile stimuli alone (black), combined tactile plus auditory stimuli (red), and combined tactile plus visual stimuli (orange) for exemplar neurons shown in **b**. **d** Comparison of the modulatory effects of various stimuli on tactile responses among individual neurons. Neurons are classified into three groups by whether SAM tones caused no change (white), inhibition (gray), or facilitation (black) in their tactile responses. Neurons are then further subdivided by whether a second stimulus (noise or light) caused no change, inhibition, or facilitation (subdivisions within each column) in tactile responses. Histograms show counts of neurons in each subdivision. Vertical bright green bars highlight categories where neurons showed the same behavior for both conditions. **e** Fraction of neurons showing same behavior (no change, inhibition, or facilitation in tactile response) for two different stimulus types. This fraction corresponds to the number of neurons that fall within the thin bright green bars in **d**. Four comparisons were performed between pairs of sounds (pale red) while three comparisons were performed between light and a sound (pale orange). The grid shows chi-squared test for proportions between each fraction to check for a significant difference. All same-modal comparisons were similar while all different-modal comparisons differed.

depends upon tactile stimulus frequency with low frequency tactile stimuli preferentially suppressed.

To exclude the possibility that the sound-driven inhibition curve is particular to the parameters of the SAM tone, we measured $r_{sound}$ at different sound modulation frequencies (Fig. 8). Inhibition in individual neurons was preserved across sound modulation frequency (Fig. 8a) and, for the population of sound-inhibited neurons in S2, $r_{sound}$ curves did not shift

(Fig. 8b–e, $p = 0.14$ for Pearson's correlation between tactile frequency at which $r_{sound} = 0.5$ and the SAM modulation frequency). Because the calculation of $r_{sound}$ is prone to error for small responses seen with low-frequency tactile stimuli (Fig. 8c), whether $r_{sound}$ is monotonic or u-shaped could not be firmly determined.

This spectral dependence of sound-modulation of tactile responses has two distinct functional consequences. First, the

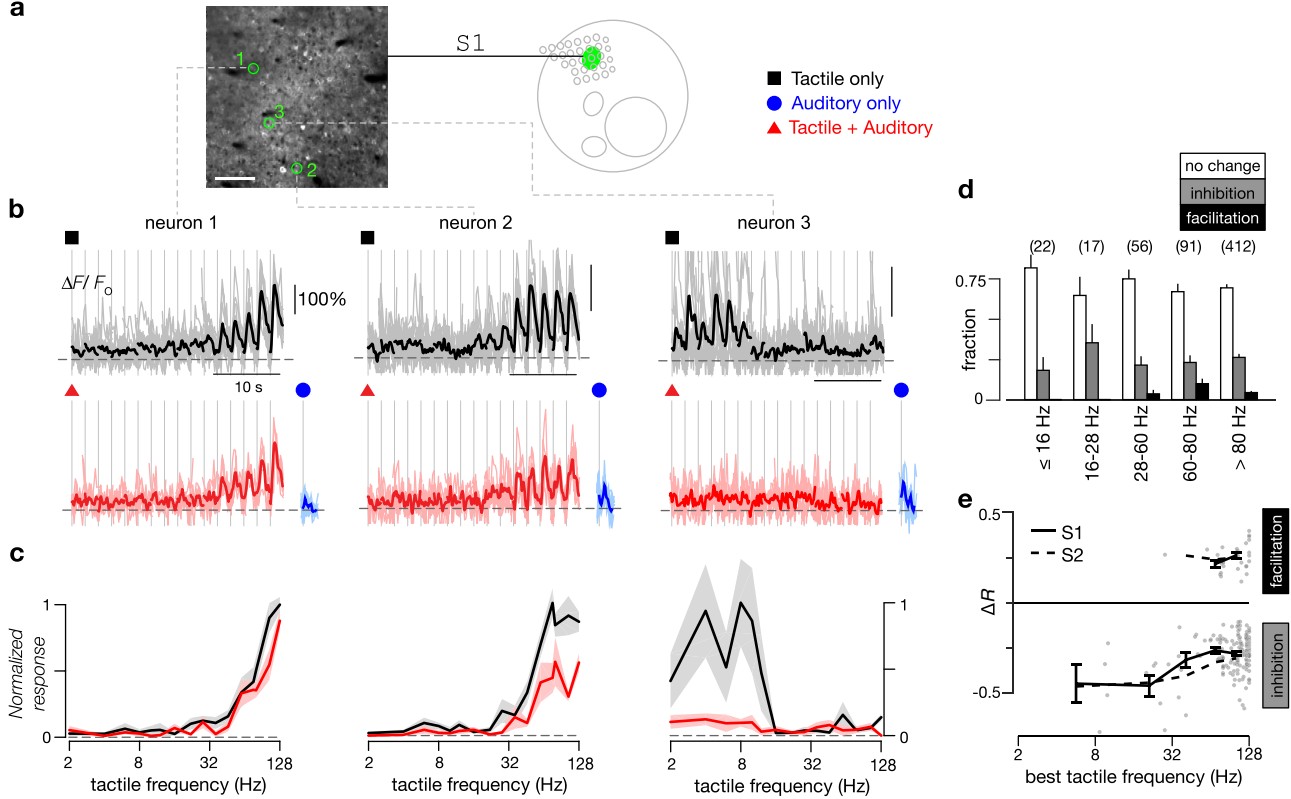

**Fig. 6 Sound-modulated response to tactile stimuli in S1. a** Baseline fluorescence image of two-photon imaging field located in S1, as illustrated in schematic map. Symbols denote different stimuli tested. Exemplar neurons to be scrutinized in panels **b**, **c** are highlighted by green circles. Scale bar: 100 μm. **b** Responses of exemplar neurons to tactile stimuli alone (black traces, 5 repeats) or to combined tactile and auditory stimuli (red traces, 5 repeats). For the combination stimulus, a concurrent auditory stimulus (10 kHz tones with 64 Hz SAM envelope at 30 dB attenuation) was added to the tactile stimuli. Blue traces show averaged responses to auditory stimulus alone. Some individual responses (light gray or red traces) are truncated to fit within the panels. **c** Frequency tuning curves of the normalized response to tactile stimuli and combined tactile plus auditory stimuli for exemplar neurons shown in **b**. **d** Fraction of neurons in S1 exhibiting either no change, a decrease (inhibition), or an increase (facilitation) when SAM tones were added to the tactile stimuli, as in Fig. 3c. Results pooled across 16 FOVs from 8 mice. **e** Percentage of change in response of S1 neurons to tactile stimuli (Δ$R$) when SAM tones were added as a function of $BTF$, shown for inhibited or facilitated neurons as described in **d**. Same format as Fig. 3d. Inhibition curve for S2 is reproduced here from Fig. 3d as a dashed line.

preferential sound-driven inhibition at low tactile frequencies results in a rightward shift in the overall tactile tuning relation for individual neurons. This effect manifests as a positive change in $BTF$ for a majority of neurons (Fig. 7e). Second, the overall inhibition of tactile responses is higher for neurons with low $BTF$ (Fig. 7f).

**Touch inhibits responses of sound-selective neurons in S2.** A fraction of neurons in S2 that were unresponsive to tactile stimuli were instead responsive to auditory stimuli (1.2% of neurons in S2, 161 neurons across 67 FOVs); for reference, 0.3% of neurons in S1 were sound-selective (12 neurons across 16 FOVs), which was significantly less than in S2 ($p < 1e-5$ by Chi-square proportion test). We aimed to determine if these auditory responses were influenced by simultaneous tactile stimulation. An exemplar sound-selective neuron in S2 did not respond to tactile stimuli at any frequency (Fig. 9b, black traces) but did respond to sound alone (Fig. 9b, blue trace). When the same sound was paired with different tactile frequencies, the sound response was almost completely abolished at the highest tactile frequencies (Fig. 9b, red traces). The normalized tuning curves demonstrate the tactile frequency-dependent suppression of sound responses in this neuron (Fig. 9c). This trend was observed for the population-averaged tuning curve of sound-selective neurons in S2, with

suppression occurring for tactile frequencies above 32 Hz (Fig. 9d). Note that analysis was limited not just to sound-selective neurons (132 neurons in S2) but to neurons selective to the specific auditory stimulus used in the combined stimuli (82 neurons in S2 from 19 FOVs across 7 mice). The suppression was quantified by the response ratio, defined as $(r_{hi} - r_{lo})/(r_{hi} + r_{lo})$, where $r_{hi}$ is the average response to sounds paired with high frequency (76–128 Hz) tactile stimulation and $r_{lo}$ for pairing with low frequency (2–8 Hz) tactile stimulation. The distribution of these values for sound-selective S2 neurons (Fig. 9e) indicates that sound responses in many neurons were decreased by high frequency tactile stimuli (31 of 82 neurons met criteria for significant suppression). This suppression was not particular to the choice of SAM tone modulation frequency: the tactile frequency that suppressed the sound response to 1.5x above baseline was flat across a range of modulation frequencies (Fig. 8f, $p = 0.66$ for Pearson's correlation).

To ascertain whether this effect was specific to S2, we measured the effect of tactile stimulation on the responses of sound-selective neurons in auditory cortex. The addition of tactile stimulation at any frequency did not change the response of an exemplar neuron to sound (Fig. 9g, h) and, across the population of sound-responsive neurons in auditory cortex, neurons exhibited minimal tactile frequency dependent changes in their sound response (Fig. 9i, j). Of 132 neurons that responded to the

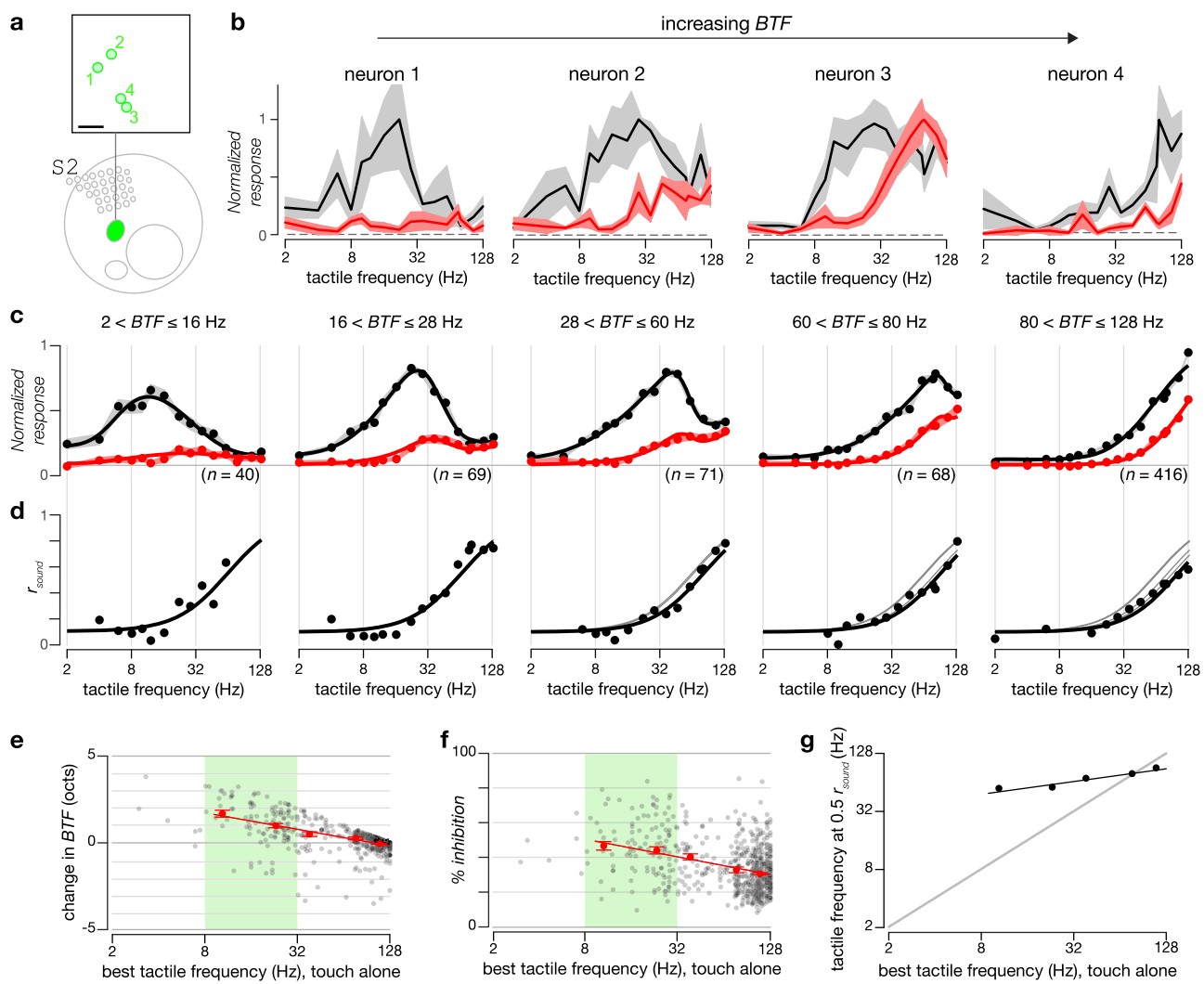

**Fig. 7 Sound-driven inhibition in S2 depends on tactile frequency. a** Locations of exemplar neurons (green circles) in two-photon imaging field located in S2, as illustrated in schematic map. Scale bar: 100 μm. **b** Frequency tuning curves of the normalized response to tactile stimuli (black traces) and combined tactile plus auditory stimuli (red traces) for exemplar neurons shown in **a**. Standard error across 6 repeats indicated by shaded gray (tactile alone) and pink (tactile plus auditory). Auditory stimulation consisted of 7.8 kHz tones with 28 Hz SAM envelope at 20 dB attenuation. **c** Population tuning curves of sound-inhibited tactile neurons in S2 grouped by their *BTF* (ranges used to categorize neurons are indicated above each tuning curve). Responses to tactile stimuli alone and tactile plus auditory stimuli are shown in black and red, respectively. Black circles indicate average responses at each individual frequency while solid lines show smooth fits to the tuning curves. Shaded regions represent standard error across neurons. Horizontal gray line indicates normalized baseline response of 0.08. **d** Sound inhibition ratio ($r_{sound}$) of neurons tuned to different *BTF*s. $r_{sound}$ is calculated by dividing the response to combined stimuli by the response to tactile stimuli at each tactile frequency after subtracting the baseline response from each. Black dots represent $r_{sound}$ at each individual tactile frequency, with frequencies that did not evoke a significant response excluded. Black lines are fitted curves. To facilitate comparison, gray lines reproduce the fitted curves of $r_{sound}$ for neurons tuned to lower tactile frequencies. **e** Change in *BTF*s of S2 neurons when sound was added (*y*-axis) as a function of *BTF* for touch alone (*x*-axis). Black dots represent individual neurons while red dots represent average change in *BTF* across all neurons within each *BTF* category. Red line is best fit to red dots. Standard errors shown in red. Green shading highlights neurons with tuning to middle frequencies (*BTF*s between 8 and 32 Hz). **f** Average percentage of inhibition of sound-inhibited tactile-selective S2 neurons when sound is presented as a function of *BTF*. Same color scheme as **e**. **g** Tactile frequencies at half-maximum ($r_{sound} = 0.5$) for neurons tuned to different tactile frequencies.

auditory stimulus used in the combined stimulus (out of a total of 267 neurons responsive to any auditory stimulus), only 11 met criteria for significant suppression, which was a significantly lower fraction than in S2 ($p < 1e-6$, Chi-square proportion test). Thus, tactile suppression of auditory responses in S2 is not unique but is much stronger than in auditory cortex.

**Spatial layout of functional cell types in S2**. Given that auditory responses of sound-selective neurons in S2 are often robustly inhibited at high tactile frequencies, which matches the inflection point of sound-driven suppression of tactile-selective neurons in

S2 (Fig. 8e, f), we posit a mutually inhibitory local circuit within S2 that serves to regulate the strength of tactile-auditory inhibition (Fig. 10a). Under this model, sound-selective neurons inhibit tactile responses of touch-selective neurons across the spectrum of whisker deflection frequencies. At low frequencies, this inhibition is nearly total, but at high frequencies the stronger activation of touch-selective neurons as a population and their reciprocal inhibition of the sound-selective population of neurons is enough to tilt the balance towards tactile responses.

We asked whether S2 exhibited any spatial clustering of neurons based on their functional properties, which could provide indirect

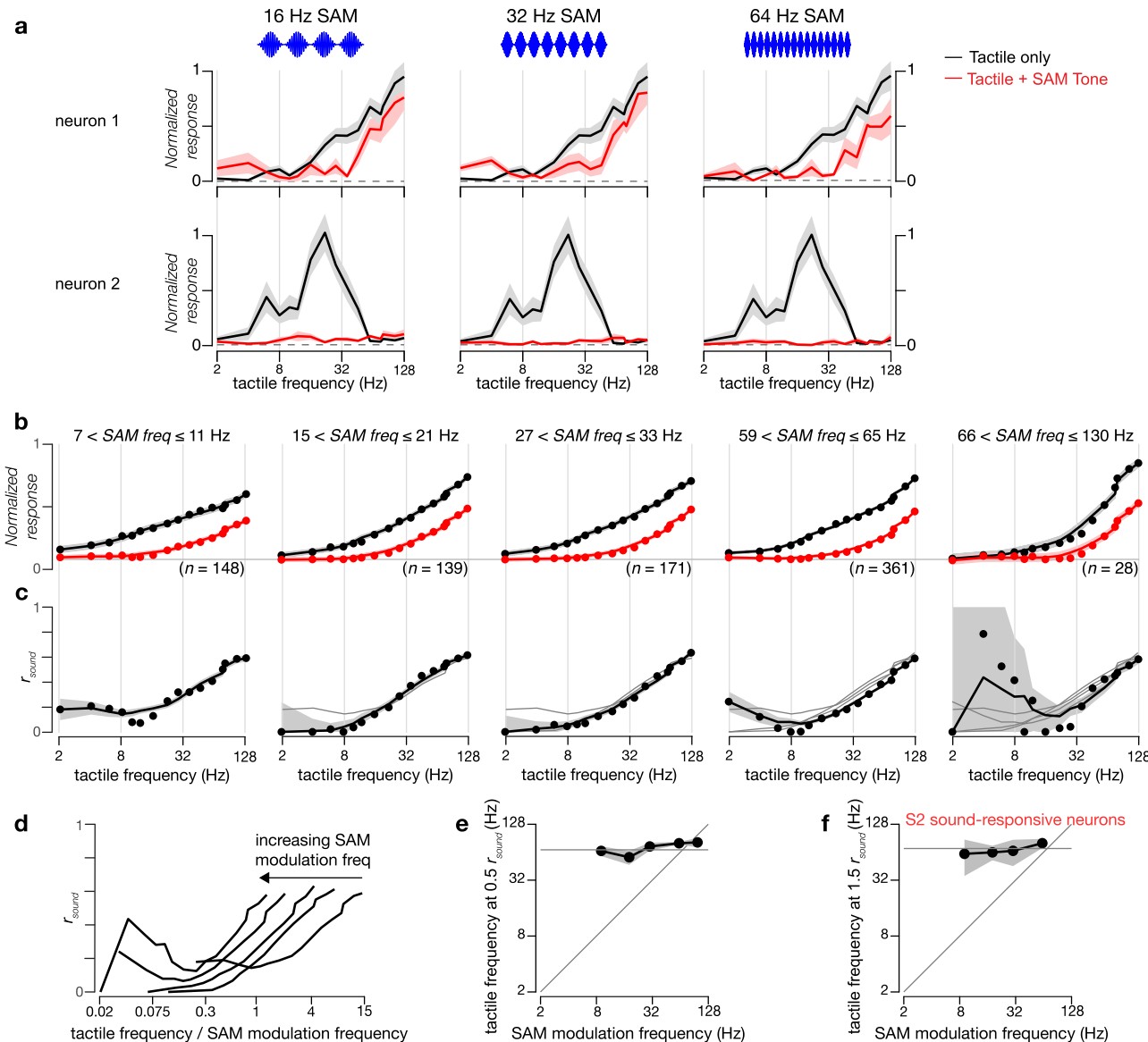

**Fig. 8 Inhibition of S2 tactile responses is invariant to sound modulation frequency. a** Frequency tuning curves of the normalized response to tactile stimuli (black traces) and combined tactile plus auditory stimuli (red traces) for two exemplar neurons at three different SAM tone modulation frequencies (16, 32, and 64 Hz). Standard error across 11 repeats indicated by shaded gray (tactile alone) and across 5 repeats in pink (tactile plus auditory). Concurrent auditory stimulation consisted of 10 kHz tones at 20 dB attenuation. **b** Population tuning curves of sound-inhibited tactile neurons in S2 grouped by SAM tone modulation frequency. Responses to tactile stimuli alone and tactile plus auditory stimuli are shown in black and red, respectively. Black circles indicate average responses at each individual frequency while solid lines show smooth fits to the tuning curves. Shaded regions represent standard error across all neurons within each specific tactile frequency domain during tactile (gray) and combined (pink) stimulation. Horizontal gray line indicates normalized baseline response of 0.08. **c** Sound inhibition ratio ($r_{sound}$) of neurons for different SAM tone modulation frequencies. $r_{sound}$ is calculated as in Fig. 7d with gray regions indicating SEM. To facilitate comparison, gray lines reproduce the fitted curves of $r_{sound}$ for neurons at lower SAM tone modulation frequencies. **d** $r_{sound}$ plotted as a function of the ratio of the tactile frequency to the SAM tone modulation frequency. **e** Tactile frequencies at half-maximum ($r_{sound} = 0.5$) for different SAM tone modulation frequencies. Bootstrap with replacement (1000 iterations) used to estimate SEM for gray region. **f** For the population of touch-inhibited sound-responsive neurons in S2, increasing tactile frequencies reduced the enhancement of response induced by sound, as in Fig. 9d. The tactile frequency at which this enhancement was reduced below 1.5 times the baseline response is here plotted as a function of SAM tone modulation frequency. The numbers of neurons for the four points are 60, 43, 22, and 55 neurons (low to high modulation frequency). Bootstrap with replacement (1000 iterations) used to estimate SEM for gray region.

evidence for a mutually inhibitory circuit. This patterning should reflect the preference of cortical neurons to connect to their nearest neighbors within 100–200 μm[29]. In particular, sound-selective neurons that were not modulated by tactile stimuli should be located further away from the C2-whisker-responsive region of S2 than their touch-modulated counterparts. To facilitate comparison of FOVs across mice, we used the local (~100 μm radius)

soma-excluded neuropil signal at each pixel in the image to form a map of tactile response strength (Fig. 10b–d). From this map, we could identify the center of the S2 whisker-responsive region (black cross, Fig. 10d), allowing us to register all S2 neurons across all FOVs (Fig. 10e). Neurons are color-coded according to whether they are touch-selective (responsive only to touch; blue circles) or sound-selective (responsive only to sound; orange and hot pink

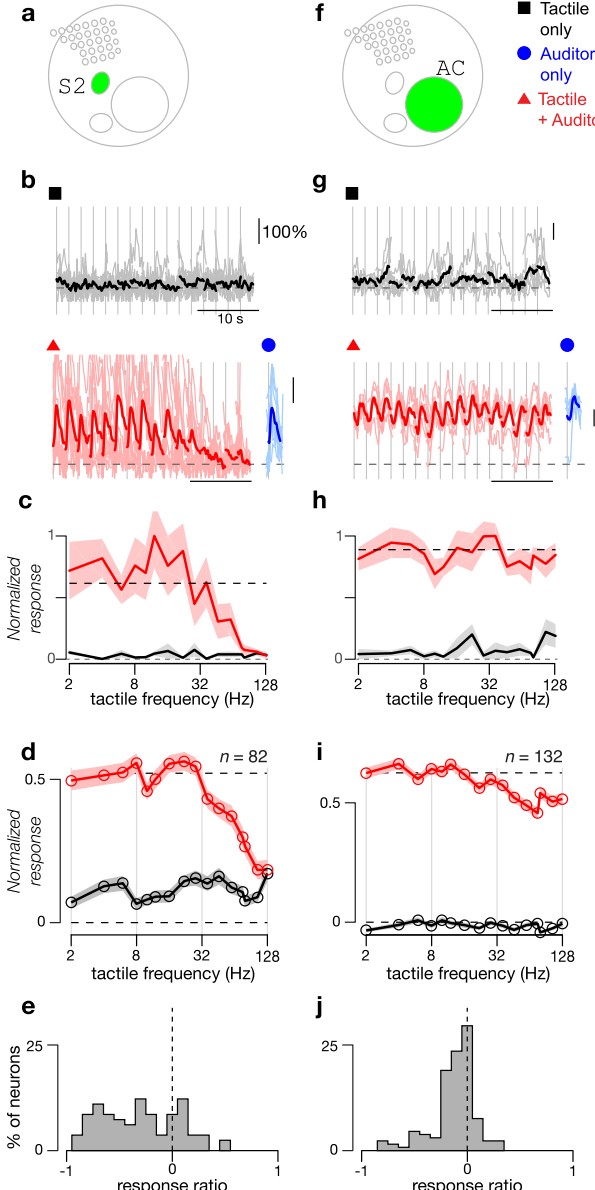

Fig. 9 Sound responses in S2 but not in auditory cortex are modulated by touch. a Schematic map shows location of S2. b Responses of exemplar sound-selective neuron to tactile stimuli alone (black, 8 repeats) or to combined tactile and auditory stimuli (red, 11 repeats). Blue trace shows average response to sound alone. Individual trials shown as light traces and response averages as dark traces. Sounds consisted of 6 kHz tones with 10 Hz SAM envelope at 20 dB attenuation. c Frequency tuning curves of the normalized response to tactile stimuli alone (black) and combined tactile plus auditory stimuli (red) for exemplar neurons shown in b. Standard error shown as shaded gray and red regions. d Averaged frequency tuning curves of the normalized response to tactile stimuli and combined tactile plus auditory stimuli for sound-selective neurons in S2 that respond to the auditory stimuli used in the combined stimuli ($n = 82$; 29 FOVs from 10 mice). Baseline firing rate was additionally subtracted. Standard error shown as shaded regions. e Distribution of response ratio at high tactile frequencies relative to low tactile frequencies for sound-selective neurons in S2 shown in c. Response ratio is calculated as $(r_{lo} - r_{hi})/(r_{lo} + r_{hi})$, where $r_{lo}$ is the response to sounds paired with low frequency (2–8 Hz) tactile stimuli and $r_{hi}$ the response to sounds paired with high frequency (76–128 Hz) tactile stimuli. f Schematic map shows location of auditory cortex (AC). g Exemplar neuron from auditory cortex. Sounds consisted of 10 kHz tones with 64 Hz SAM envelope at 30 dB attenuation. Same format as b. Each stimulus repeated 5 times. h Frequency tuning curves of the normalized response to tactile stimuli and combined tactile plus auditory stimuli for exemplar neuron shown in g. Same format as c. i Averaged frequency tuning curves of normalized response to tactile stimuli and combined tactile plus auditory stimuli for sound-responsive neurons in auditory cortex that respond to the auditory stimuli used in combined stimuli ($n = 132$). Same format as d. j Distribution of response ratio (as in e) for sound-responsive neurons in auditory cortex.

(Fig. 10g), providing indirect evidence of inhibitory connectivity between touch-inhibited sound-selective neurons and local touch-selective neurons in S2.

## Discussion

In this work, we asked how information from somatosensory and auditory inputs is integrated in the mouse neocortex. With two-photon $Ca^{2+}$ imaging, we investigated large populations of layer 2/3 neurons across somatosensory and auditory areas with single cell resolution. We found that neurons across somatosensory cortices are tuned to the frequency of tactile stimulation. The addition of concurrent sound resulted in modulation of these tactile responses in both S1 and S2, and this modulation typically manifested as a suppression of the response. Moreover, the degree of suppression depended on tactile frequency, with responses to low frequencies more inhibited than responses to high frequencies. We also identified a population of neurons in S2 responsive to sound but not to touch. Unlike in auditory cortex, sound responses of many (31 of 82) sound-selective neurons in S2 were strongly inhibited by addition of tactile stimuli at high tactile frequencies. These neurons were spatially colocalized with S2 touch-selective neurons.

The detection of the frequency of mechanical vibrations is important for animals to discern surface texture and to handle tools[30,31], and tuning to spectral frequency in the somatosensory system can encode texture information[32]. In our study, the presence of well-tuned neurons in both S1 and S2 supports the notion that tactile frequency tuning may be a general organizational feature for mouse tactile sensation. The higher proportion of neurons with tuning to lower tactile frequencies in S2 than in S1 may reflect differences in thalamocortical inputs to the two regions. S1 receives strong thalamic drive from the ventral posterior medial nucleus (VPM), while S2 receives a larger share of its thalamocortical input from the posterior medial nucleus (POm)[33,34]. Interestingly, although both POm and VPM cells

circles). Neurons responsive to both touch alone and sound alone ($n = 43$) were excluded from this analysis. We found that touch-selective neurons were more clustered (assessed by distance to the center of the S2 whisker-responsive region) than the average neuron in the field (blue and black curves, Fig. 10f). Sound-selective neurons that were not inhibited by touch exhibited a similar distribution to all neurons (orange and black curves, Fig. 10f), indicating that they were equally distributed across a typical FOV and perhaps even partially excluded from the whisker-responsive center of S2. However, when we examined the distribution of sound-selective neurons that showed significant suppression of responses by tactile stimuli, as in Fig. 9b, c, the distribution of distance from the S2 core now closely mirrored that of touch-selective neurons (hot pink and blue curves, Fig. 10f). These results indicate that both touch-selective neurons and touch-inhibited sound-selective neurons are embedded together in a specific region of S2, hinting at a local circuit linking these two functional subtypes. Further corroborating this model, spontaneous correlations between touch-inhibited sound-selective neurons with tactile-responsive neurons was lower than the correlations between non-touch-inhibited sound-selective neurons with tactile-responsive neurons

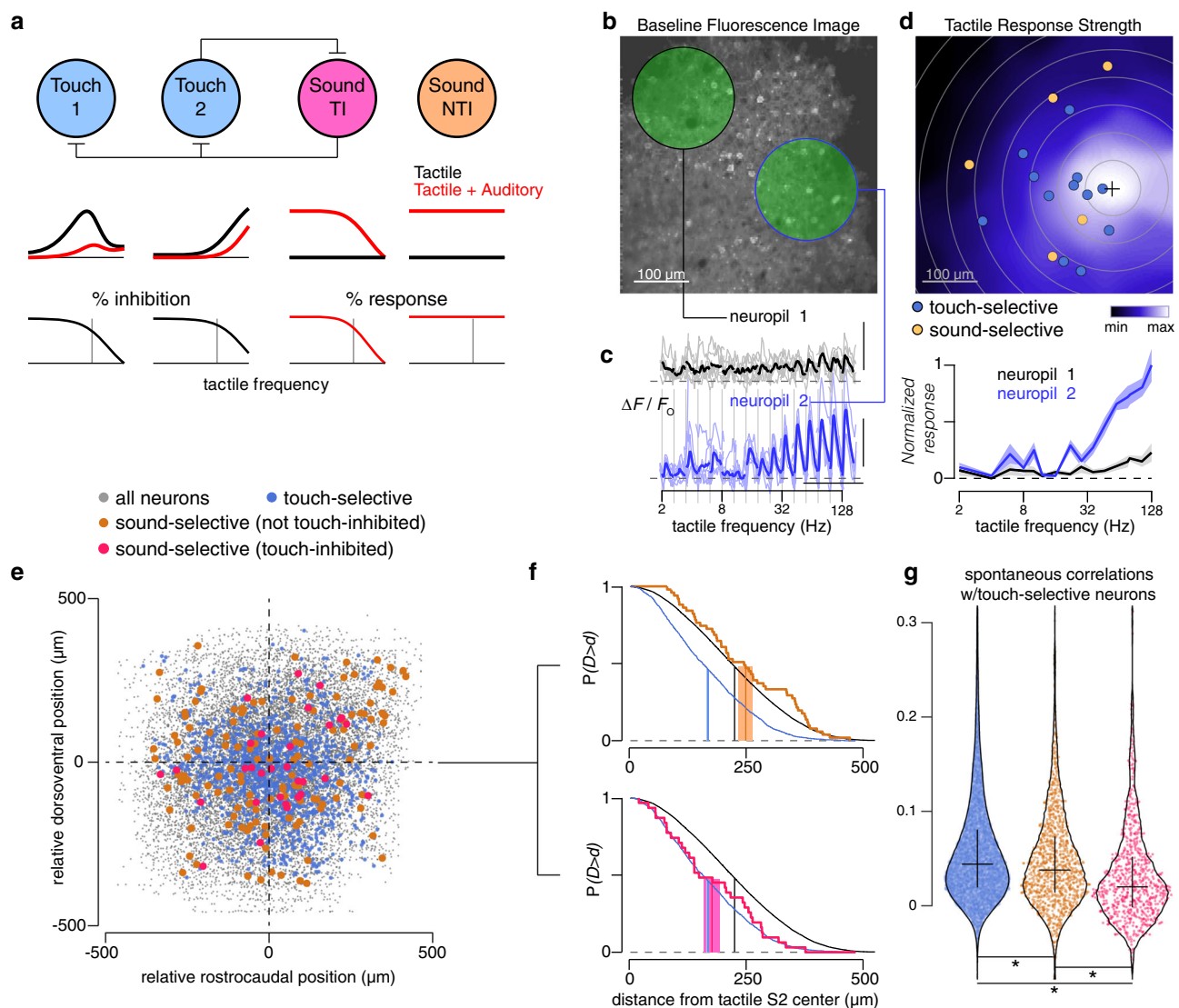

**Fig. 10 Spatial layout of multisensory neurons in S2. a** Model of multisensory interactions in S2. Sound-responsive and touch-responsive neurons are connected via reciprocal inhibition. Touch-inhibited sound-responsive neurons ('Sound TI') connect with touch-responsive neurons while non-touch-inhibited sound-responsive neurons ('Sound NTI') do not. Tactile tuning curve with or without concurrent sound are shown. Percent inhibition of tactile response induced by concurrent auditory stimulation (black traces) or of auditory response induced by concurrent tactile stimulation (red traces) are shown underneath. Vertical gray line represents tactile frequency of 32 Hz. **b** Baseline fluorescence image of two-photon imaging field in S2. Green circles highlight two neuropil regions. Cell bodies were excluded when calculating neuropil signals. **c** Fluorescence responses and frequency tuning curves for two neuropil regions. Averaged across 6 repeats. Scale bars: 10 s, 25%. **d** Map of tactile response strength for the FOV in panel **b**. Response strength was calculated by taking the total area under the frequency tuning curve for a neuropil region of interest centered on that pixel. Black cross marks the pixel with maximum response. Touch-selective neurons (blue) and sound-selective neurons (yellow) are indicated as well. **e** Registered map of all S2 neurons. Individual FOVs were centered using the peak of their neuropil tactile response strength map. **f** Tail distributions of distance from tactile response strength center for different neuronal types. All neurons (black) and touch-selective neurons (blue) are shown on both plots. Orange trace shows NTI sound-selective neurons while hot pink trace shows TI sound-selective neurons. Vertical lines indicate means while shaded regions represent ±SEM. **g** Pairwise correlations during periods of spontaneous activity (absence of both tactile and auditory stimulation) between touch-selective neurons and the three groups indicated by color (blue: other touch-selective neurons; orange: non-touch-inhibited sound-selective neurons; hot pink: touch-inhibited sound-selective neurons). These groups contained 19743, 1164, and 748 neuron pairs, respectively. Pairs within 15 μm were excluded. All groups are significantly different from each other (unpaired *t*-test, *p < 1e-6). Violin plot generated by 7-point smoothed density function with 100 bins (0.5% of neurons outside range are excluded from plot). Cross indicates 25-50-75 quartiles.

show adaptation, causing decreased response amplitude under high frequency stimulation, POm cells exhibit earlier adaptation than VPM cells[35] and as a result are tuned to lower frequencies than VPM cells. Thus, the tuning properties of neurons in S2 may be inherited from the response properties of thalamic neurons, although it could also reflect longer temporal integration windows in higher areas of cortex[36].

We found that the addition of an auditory stimulus modulated tactile responses in both S1 and S2, consistent with the sound-driven hyperpolarizing currents previously observed in mouse S1[37]. This modulation has three notable features: (1) Although a similar proportion of neurons in both S1 and S2 were facilitated by sound, more neurons in S2 were inhibited than in S1 (Figs. 3d and 5d). (2) Inhibition of neurons tuned to low tactile frequencies in

both S1 and S2 was more severe than inhibition of neurons to high tactile frequencies in the same regions (Figs. 3e and 5e). (3) Sound-driven suppression in S2 is tactile frequency dependent, with stronger inhibition occurring at lower tactile frequencies (Figs. 7 and 8). Previous studies in human and non-human primates have revealed that multimodal integration improves detection of events in the environment[3,38,39]. The optimal integration of competing sensory cues involves dominance of the most reliable sensory cue to minimize variance in the estimation of the true stimulus[3]. This evaluation of reliability between different sensory cues is a dynamic process, with the weight or value of each stimulus modality being continuously updated[39]. Low frequency tactile stimulation is potentially less salient of a signal than high frequency tactile stimulation, since it comprises lower velocity whisker motions. Indeed, we observed more suppression of tactile responses at lower tactile stimulus frequencies than at high frequencies (Figs. 7 and 8), indicating that auditory responses are more dominant when tactile stimuli are weak. This result is consistent with the prior observation that, during optimal multimodal integration, the more reliable stimulus modality dominates the response[40]. On the other hand, this frequency-dependent integration is complementary to "inverse effectiveness," where multimodal integration is largest for weak multimodal stimuli near threshold and decreases with increasing stimulus intensity, as has been reported in the superior colliculus[41,42].

Sound-touch modulations may involve more than just direct interactions between the unimodal stimuli themselves. Attention, arousal, motor behavior, and hidden internal states can be influenced by sensory stimuli and they, in turn, can influence the response to a sensory stimulus[23,24,43–45]. Indeed, multisensory integration, if relevant to behavior, should be associated with a change in the internal state of the animal. Pointing towards this complex interplay of stimuli and internal states, we found that locomotive behavior, while able to influence sensory responses, could not explain sound-driven inhibition of tactile responses on its own (Fig. 4). To untangle these potentially complex interconnections, the underlying cellular and network mechanisms that mediate these interactions need to be uncovered. While our present work focused on neurons in layers 2/3, a fruitful locus of study would be layers 1 and 6, where crossmodal[46] and neuromodulatory[47] inputs are known to be stronger and may thus gate sensory inputs and mediate attentional effects.

Previously, it was believed that multimodal influences on activities within classically defined unimodal areas are mediated by feedback from multisensory integration in higher-order cortical regions[48,49]. However, human studies using event-related potentials (ERPs) suggest that these multimodal influences may also be carried in the feedforward inputs coming from subcortical regions to unimodal regions[48,50,51]. In the present study, we identified a small (1.2%) population of sound-selective neurons within S2 itself. Although prior studies have shown non-matching neurons in primary cortices that respond solely to other sensory modality inputs[52], the sound-selective neurons we found may play a special computational role in multimodal integration. The sound-driven responses in these neurons were strongly suppressed at high tactile frequencies (Fig. 9a–e), and those inhibited by tactile stimuli neurons are clustered near the center of the whisker-responsive region of S2 (Fig. 10), similar to the spatial organization of non-matching neurons seen in other studies[51,52]. The existence of touch-inhibited sound-selective neurons in S2 indicates that they may play a role in local sound-driven suppression observed in tactile-selective neurons of S2. This winner-take-all circuit (Fig. 10a) could dynamically select a stimulus modality at each moment and, under the right conditions, would be consistent with divisive normalization, a model that has been proposed as a driving force behind multimodal interactions[53–55].

## Methods

**Animal surgery and general procedures.** All animal procedures were approved by the Johns Hopkins Institutional Animal Care and Use Committee. Imaging experiments used transgenic mice 10–22 weeks of age expressing the genetically encoded $Ca^{2+}$ indicator GCaMP6s[56] under a pan-neuronal promoter. Most experiments used GP4.12 Thy1-GCaMP6s mice (JAX No. 025776)[12]. A small subset of tetO-GCaMP6s mice (JAX No. 024742) were also used for imaging[14]. Finally, lox-GCaMP6s mice (JAX No. 024106)[13] were crossed with Syn1-Cre mice (JAX No. 003966)[57], resulting in lox-GCaMP6s-Syn1-Cre mice. In total, 18 males and 5 females were imaged for this study. Among them, 17 were GP4.12 Thy1-GCaMP6s mice, 3 were tetO-GCaMP6s mice, and 3 were lox-GCaMP6s-Syn1-Cre mice.

For surgery to implant headposts and chronic imaging windows, general anesthesia was provided by 1–1.5% isoflurane in 0.5 L/min $O_2$. Body temperature was monitored through a rectal probe and maintained at 37 °C with a heating pad. Lidocaine (20 mg/ml) was injected at the incision site for local anesthesia while carprofen (5 mg per kg, i.m.) and dexamethasone (5 mg per kg, i.m.) were given for pain and inflammation. Normal saline (0.5 ml, i.p.) was given to avoid dehydration. Skin and muscle over the left hemisphere were removed to expose skull overlying auditory and somatosensory cortices and a custom headpost was attached to the skull using UV-cured primer (Kerr OptiBond) and dental cement (Heraeus Charisma A1). A 4–5 mm craniotomy spanning both auditory and somatosensory cortices was performed. After removing the skull, the craniotomy was filled with a transparent silicone gel (3–4680, Dow Corning, Midland, MI)[58] to help maintain the stability and clarity of the chronic window. Next a 4 or 5 mm (No. 0 or 1) cover glass was secured with dental cement above the window. Mice were given buprenorphine (0.1 to 0.5 mg per kg, s.c.) for pain and allowed to recover from anesthesia on a warmed pad. After surgery, mice were periodically given buprenorphine for pain along with carprofen and dexamethasone. Mice were given 10–14 days to recover before imaging sessions were started. Location of the C2 barrel column in S1 was estimated by stereotaxic measurement of its expected location relative to lambda (1.3 mm posterior and 3.5 mm lateral). This location matched the most medial of the three regions responsive to tactile stimulation as seen with functional widefield imaging (Fig. 1).

In a subset of experiments, to examine whether sound-driven inhibition was dependent on active whisking, the facial nerve was transected. The buccal branch of the facial nerve innervates the mystacial pad musculature[59], which in turn controls whisker movement[60]. General anesthesia was achieved with isoflurane (1.5% in 0.5 L/min $O_2$) and body temperature was maintained at 37 °C. Local anesthetic (lidocaine, 20 mg/ml) was injected at the incision site, which was 1–2 mm caudal of the E1 whisker. The surrounding tissue was dissected to reveal the buccal branch of the facial nerve at the junction of the ramus superior buccolabialis and ramus inferior buccolabialis[25,61]. Micro scissors were used to transect the nerve just anterior to the junction point and proximal to entering the mystacial pad. Skin closure was achieved with nonabsorbable suture (4–0 Prolene) and Vetbond (3 M). Mice were allowed 2–3 days to recover. The absence of active whisking ipsilateral to the site of the transection was used to assess the success of the surgery. Imaging was performed as described below until active whisking returned, typically 2 weeks after the initial cut.

**Experimental setup.** During imaging sessions, all mice were awake and head-rotated ~45° to make the imaging plane perpendicular to the microscope objective. Mice were first adapted to running on a custom-built treadmill while head-fixed for 1–2 session of 20 to 40 min each. The treadmill consisted of a spherical ball with a single axis of rotation. Once habituated, imaging sessions began. Mice were able to tolerate head fixation for up to 2 or 3 h. Running speed was recorded with an optical encoder (E5-2500-188-IE-S-H-D-B, US Digital). For whisker stimulation, all whiskers of the right whisker pad were trimmed except for the C2 whisker, facilitating isolated stimulation of a single whisker.

**Imaging.** For widefield imaging, a white light source (LED Engin LZ1-10CW00) was used for illumination. Illumination light and fluorescence signals were filtered through a GFP filter cube (460/50 excitation, 540/50 emission). A 4× 0.13 NA objective (Olympus) and a Photometrics Evolve 512 Delta camera were used to collect the emitted light. The field of view was roughly $5.5 \times 5.5$ mm². The frame rate was 20 Hz.

Two-photon $Ca^{2+}$ imaging was performed with an Ultima system (Prairie Technologies) built on an Olympus BX61W1 microscope. A mode-locked laser (Coherent Chameleon XR Ti:Sapphire) tuned to 950 nm was raster scanned at 5 Hz for excitation while emitted GCaMP6s fluorescence was collected through a green filter (525/70 nm). Laser power at the sample was 20–80 mW. Dwell time was set to 2 µs. To increase imaging speed, resolution along the y-axis was reduced by a factor of 4. The final pixel size was $0.9 \times 3.6$ µm. A 20× 1.0 NA objective (Olympus) was used to yield a $465 \times 465$ µm² field of view. Imaging depths were between 150–350 µm.

**Tactile stimuli.** To stimulate the target whisker, a piezoelectric bending actuator (Q220-A4-203YB, Piezo systems, Inc., MA) moving in the rostral-caudal direction was attached to a 2 cm sewing needle and the target whisker was passed through the eye of the needle at a distance of 2 mm from the face. The needle coupled the whisker to the piezo movement. The shape of the eye of the needle allowed the

whisker some freedom to move in the medial-lateral direction but not in the rostral-caudal direction. Positioning the needle close to the base of the whisker reduced flexion of the whisker during stimulation, thus minimizing nonlinear mechanical transformations of the sinusoidal driving signal. To facilitate precision in the positioning of the needle, we attached the piezo to a micro-manipulator (MT-XY Compact Dovetail Linear Stage, Newport Corporation). The piezo itself was driven by a voltage driver (MDT694B, Thorlabs) with a 10× gain. Thus, the 0–10 V signal delivered by the DAQ card was expanded to a range of 0–100 V to drive the piezo.

A single tactile stimulus consisted of a 500 ms sinusoidal wave. One tactile trial consisted of a set of sixteen individual stimuli with onsets spaced by 2 s. Frequency of each sinusoidal stimulus was randomized between 2 to 128 Hz, spaced logarithmically. The amplitude of piezo deflection was 0.6 mm. With the needle 2 mm away from skin surface, a 2 Hz stimulus was equivalent to a mean angular speed of 66.8°/sec, calculated as $\tan^{-1}(0.6 \text{ mm}/2 \text{ mm}) \times 2 \text{ Hz} \times 2$ (one forward and one backward deflection per cycle). Thus, mean angular speed of our tactile stimuli is calculated from frequency as 33.4° per sec × (frequency), where frequency is given in Hz. In this work, we report the stimulus parameters in frequency. The piezo was calibrated monthly by imaging piezo movement to measure maximum deflection displacement and angular velocity. Once angular velocity dropped by more than 20%, the piezo was replaced.

An important consideration in this work was whether the delivery of tactile stimulation with the piezo generated any audible noise within the hearing range of mice. We performed a noise calibration while driving the piezo at frequencies ranging from 2 to 256 Hz. Sound was recorded with a Sokolich probe placed directly next to the piezo. We found that no noise audible to mice (across the mouse hearing range of 3–100 kHz) was generated as long as the frequency of piezo deflection was kept lower than 140 Hz. Above 140 Hz, though, audible sounds were generated (Supplementary Fig. 1d, e). Thus, all stimuli used in this study were limited to a maximum of 128 Hz. In support of this assertion, only 2 out of 2036 neurons recorded in auditory cortex passed criteria for a response to stimulation by the piezo alone. To test the opposite, whether acoustic stimuli could cause any vibration of the piezo, we imaged the piezo through the microscope objective at a frame rate of 363 Hz while playing sounds at the maximum amplitude possible with our speakers. We found no detectable movement of the piezo under imaging.

**Auditory stimuli.** The imaging set-up was located in a sound-attenuated room (Acoustical Solutions, AudioSeal ABSC-25) with noisy equipment placed outside the room. Noise calibrations showing ambient noise to be outside the mouse hearing range have been previously published[15]. Sound stimuli were delivered through a free-field speaker (LCY K100, Ying Tai Corporation) located 15 cm away from the right ear.

The main stimuli used were sinusoidal amplitude modulated (SAM) tones, which were composed of a pure tone (the carrier frequency) with a low frequency amplitude modulation. To search for sound responses, SAM tones were delivered over a wide range of carrier frequencies (3 to 96 kHz, 16 frequencies) or amplitude modulation frequencies (2 to 256 Hz, 8 frequencies). The duration of each stimulus was 500 ms (Supplementary Fig. 1b). We tested a variety of additional sounds in a subset of experiments, including pure tones and broadband noise (Fig. 5). Pure tones were gated by 5 ms squared cosine ramps. White noise was bandlimited between 3 to 96 kHz and, if indicated, was amplitude modulated by a sinusoidal envelope (SAM Noise). Neurons were categorized as sound-responsive if they responded to any of the auditory stimuli tested. All the stimuli were delivered in a random order and with a 2 s interval between onsets. Stimuli were played at −40 to −20 dB attenuation, corresponding to ~60–80 dB SPL using our speaker. Sound levels were chosen by looking for reliable responses in auditory cortical areas as visualized by widefield fluorescence imaging.

**Combined stimuli.** Combined stimuli consisted of simultaneous presentation of the auditory and tactile stimuli described above. The tactile stimuli were sixteen tactile deflections with different frequencies ranging from 2–128 Hz while a fixed auditory stimulus was presented concurrently with each of the sixteen tactile stimuli. For the auditory stimuli, SAM tones were used for every FOV, with the particular carrier and modulation frequencies chosen such that the stimulus had elicited a robust response in auditory regions during widefield fluorescence imaging (typically between 6–12 kHz for the carrier frequency and 8–64 Hz for the modulation frequency). As the choice of sound could have a bearing on the observed multisensory effects in somatosensory cortex, we also tested pure tones and broadband noise (with or without amplitude modulation) in some experiments (Fig. 5). Each stimulus lasted 500 ms and the tactile and auditory stimuli were co-initiated and co-terminated (Supplementary Fig. 1c). Stimulus order was randomized with a 2 s interval between the onset of each presentation.

**Stimulus order.** In general, we interleaved blocks of each stimulus condition, for example one block of tactile alone (consisting of 16 piezo frequencies) followed by one block of tactile + auditory (16 piezo frequencies in combination with one fixed sound stimulus). Within each block, the stimuli (such as the 16 piezo frequencies) were always randomized, stimuli lasted 500 ms, and intervals were 2 s. The blocks were interleaved in a semi-random order as chosen by the experimenter, with the

goal being to spread each stimulus block across the duration of the imaging session to minimize the effects of habituation and bleaching. At least 4 repeats were attained for each condition, although typically we aimed for 5 to 10 repeats. Whenever responses are shown, all individual trials are included.

**Widefield imaging analysis.** For widefield imaging, a structured sparse encoding algorithm was used to detect the baseline fluorescence $F_0$[15,62,63], from which $\Delta F/F_0 = (F - F_0)/F_0$ was calculated for each pixel. To facilitate comparison of changes across different regions of cortex, plotted time series data (Fig. 1b, c, e, f) were zeroed at the start of each stimulus.

**Two-photon imaging analysis.** We first performed a frame-by-frame image registration to correct for drift in our imaging field[64]. Assisted by a structured sparse coding algorithm[62], we then performed a semiautomatic segmentation of our movies acquired from a two-photon imaging session to acquire regions-of-interest (ROIs) corresponding to individual neurons. From each ROI, we generated brightness-over-times (BOTs). This process included neuropil subtraction, where we subtracted the signal from pixels within 75 μm of each cell (but excluding pixels containing other cell bodies). Using a low-pass filter that excluded large positive deflections (putative transients), a smooth baseline ($F_0$) was fit to each BOT, allowing for calculation of a normalized fluorescence signal $\Delta F/F_0$. To increase the temporal accuracy of our $Ca^{2+}$ signal and to decrease the influence of noise, we employed a non-negative deconvolution method with a time constant of 0.7 s to generate a correlate of spike probability[65], which was then thresholded to estimate events per time bin, which loosely corresponds to the frequency of spikes or firing rates for a given neuron[66–71]. Events per time bin was used for constructing tuning curves and other measures as described in the section of "Response Analysis." Parameters for choosing neurons in the segmentation algorithm were purposefully set to be sensitive so that no responsive neurons would be missed; conversely, many ROIs correspond to relatively inactive neurons or false detections. These ROIs would not reach significance for being responsive and thus do not alter any of the conclusions or analysis in this work other than the potential for inflating the number of nonresponsive neurons.

**Registration.** For most mice, the imaging window was initially mapped under two-photon imaging by tiling FOVs across the entire visible window and at multiple depths. Two-photon imaging fields were registered to widefield imaging fields by using vasculature landmarks. Information about the location of auditory (primary auditory cortex, anterior auditory field, and secondary auditory cortex) and somatosensory cortices (S1, S2, and ISF) obtained from widefield imaging was then used to identify the cortical identity of each two-photon imaging field.

**Response analysis.** Response analysis includes five major parts: (1) event rate normalization; (2) quality measure of responsiveness; (3) best frequency extraction; (4) construction of tuning curves; (5) calculation of sound-driven response change.

For event rate normalization, we used a 600 ms period after the start of a stimulus as our response window. The averaged baseline-corrected signal during that period yielded the event rate or response for each individual stimulus.

To judge whether a neuron is responsive to a given stimulus modality, we developed a metric that combines 3 measures: (1) the $p$-value of 1-way ANOVA to quantify whether there is a significant effect of stimulus frequency on the response, which indicates a neuron is tuned to stimulus frequency; (2) the total area under the tuning curve, a measure of driven rate; (3) the ratio of activity during the response window to activity outside the response window (Supplementary Fig. 2a, b). Each quantity is passed through a saturating nonlinearity and summed to yield a unitless measure of responsiveness (termed 'score' in Supplementary Fig. 2). This ad hoc metric is thresholded to categorize neurons as responsive or not responsive to a particular stimulus type. Individual neurons were scrutinized to test for the adequacy of this approach and the threshold chosen to best match our manual characterization of neurons. We chose this approach because any single measure was inadequate in capturing whether neurons were responsive. For example, measure 1 (1-way ANOVA) does not account for magnitude differences, so a strongly-responsive, broadly-tuned neuron may have an insignificant $p$-value (Supplementary Fig. 2b, neuron (1) while an unresponsive neuron with incomplete neuropil subtraction may have a significant $p$-value (Supplementary Fig. 2b, neuron (2). For further analysis, if the neuron responded only to tactile stimuli, we classified it as a tactile-selective neuron. Similarly, if the neuron only responded to any of the sound stimuli we tested, we classified it as a sound-selective neuron. Using a shuffle test applied to neurons from our population (100 shuffles of the times of stimulus presentation), only 1 in 4832 iterations passed criteria for touch selectivity and 1 in 14,980 for sound selectivity. We did find that sound-selective S2 neurons, while not passing criteria individually for being responsive to tactile stimuli, did on average show a small level of activity in response to tactile stimuli alone (black curve, Fig. 9d). This response was weak (~5× smaller than the sound response) and inconsistent, as confirmed by manual inspection of the 82 individual tuning curves, and thus may reflect weak and inconsistent tactile-related activity in these neurons.

A measure of best tactile frequency (BTF) was used to categorize neurons (Supplementary Fig. 2c). We used the average of two different approaches to

measure best frequency: (1) the weighted average of all significant responses; (2) the frequency that yielded the maximal response. We chose to optimize for robustness as each of these standard measures on their own is susceptible to pitfalls. The first approach has a slight bias, as neurons tuned to high frequencies will be biased by the weighted mean towards lower frequencies. The second approach can be unreliable for noisy data. Nonetheless, the two measures are highly correlated ($r = 0.913$, Pearson's correlation, Supplementary Fig. 2c), and BTFs in over 97% of responsive neurons were less than 1 octave apart by the two measures. Neurons with best frequencies less than or equal to 60 Hz are categorized as low-*BTF* neurons while neurons with best frequencies greater than 60 Hz are categorized as high-*BTF* neurons.

To construct tuning curves, we used two different strategies. For characterizing tuning properties of tactile-selective neurons in S1 and S2 (Fig. 2), we normalized the response for different tactile frequencies to the maximum response among all the tactile frequencies. For characterizing the sound modulation effect of response to tactile stimuli, we normalized the response to the maximum response of either tactile stimuli or tactile plus auditory stimuli. And, for experiments where we used a third modality (visual light stimulation in Fig. 5), we normalized the response to the maximum response across the three stimuli (tactile stimuli, tactile stimuli with concurrent sound stimuli, and tactile stimuli with concurrent visual stimuli).

To quantify the sound-driven response change of tactile stimuli, we calculated the ratio of response change ($\Delta R$) as $\Delta R = \sum (R_{T+S} - R_T)/\sum (R_{T+S} + R_T)$, where $R_T$ is the response to tactile stimuli alone, $R_{T+S}$ the response to combined tactile plus auditory stimuli, and the summations are over all tactile frequencies tested (2 to 128 Hz).

For the registration of S2 imaging fields shown in Fig. 10, we first formed smoothed neuropil maps by taking a region centered on each pixel and calculating a cell-excluded average of the signal within 100 μm of that pixel. This neuropil BOT signal was processed similarly to the neuronal BOT signal as described above, with the only difference being that neuropil subtraction was not performed on the neuropil signal. The amplitude of the whisker-induced response as quantified by summing the neuropil $\Delta F/F_0$ over a 600 ms window for each stimulus was used to generate an image of tactile response strength. The peak of this map was nominally labeled as the center of the S2 whisker region and used to register FOVs across experiments.

The generalized linear model (GLM) was used to model the response of individual neurons on a single trial basis. The output was taken as the mean response over a 600 ms window for each stimulus, as described above, and then normalized so that the maximum response at any time is equal to one. Dependent variables consisted of tactile stimulus frequency (binned into four separate frequency channels), the presence of sound, and the running velocity. Indicator functions were used for stimulus frequency and the presence of sound, while running velocity was passed through a logarithmic function: $\log_2(6^*v+1)/10$, such that the range would fall between 0 to 1 ($v$ never went higher than 170.5 cm/s in our data). Velocity was calculated based on the ball diameter of 6.75 cm and 2500 counts per rotation on the optical encoder. A velocity of 50 cm per s resulted in a value of ~0.8234 fed into the GLM, while a velocity of 0 cm per s gave a value of 0. This transformation function was chosen as it yielded the best model fits for velocity modulated neurons in our population. The logit function was used as the link function, and the MATLAB function glmfit.m was used to perform the fitting. Resulting coefficients were then z-scored by dividing by their associated standard errors. Trials were included for tactile stimulation alone, tactile stimulation with SAM tones, SAM tones alone, and, if performed, "silent" trials with no tactile or auditory stimuli.

**Statistics and reproducibility**. Sample sizes were chosen based on the consistency of the observed results. Data were excluded if the imaging quality was poor as judged by brightness of the fluorescence signal, ability to observe active transients in individual cells, and general optical sharpness of the two-photon imaging field. Replication was only indirectly tested by using large sample sizes or by varying experimental conditions and observing the results. Individual statistical tests are described where referenced in the manuscript.

**Reporting summary**. Further information on research design is available in the Nature Research Reporting Summary linked to this article.

## Data availability
Data is available upon request from the corresponding author. Image files are stored in tiff format and processed data are stored as MATLAB data structures.

## Code availability
Code (written for MATLAB R2014A) is available upon request from the corresponding author.

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

## Acknowledgements

We thank David T. Yue for conceiving of this project and for providing the scientific foundation for this work. We also wish to thank Eric D. Young and Dwight E. Bergles for comments on the manuscript, Benjamin D. Haeffele for experimental set-up and guidance on image processing, Gordon F. Tomaselli for guidance and support, Jeffrey M. Yau and Sascha du Lac for valuable discussions, Ingie Hong for assistance with assembly of the head-fixed awake preparation, Travis Babola for assistance with computer hardware, Terry Shelley for fabrication of stereotaxic equipment, Loren Looger and the GENIE project for supplying transgenic GCaMP6s mice, and members of the O'Connor lab and the Calcium Signals Lab for helpful discussions. This work was supported by grants from the Kleberg Foundation, the Whitehall Foundation, the Klingenstein Fund, and the NIH (NIMH R01MH065531, NINDS R01NS073874, NINDS P30NS050274, NINDS R01NS089652).

## Author contributions

M.Z. performed all experiments and collected all data with assistance from S.E.K. and J.B.I. All authors contributed to experimental design. D.H.O. and J.B.I. supervised the project. M.Z., M.B.J. and J.B.I. processed and analyzed the data with assistance from D.H.O. All authors contributed to the writing and editing of the manuscript with M.Z. and M.B.J. assembling the initial draft and M.Z., D.H.O. and J.B.I. revising the final version.

## Competing interests

The authors declare no competing interests.
