## [Peer Review File · Communications Biology]

Editorial Note: *This manuscript has been previously reviewed at another Nature Research journal. This document only contains reviewer comments and rebuttal letters for versions considered at Communications Biology.*

REVIEWERS' COMMENTS:

Reviewer #2 (Remarks to the Author):

The authors have satisfactorily addressed my questions, I recommend this paper for publication in Communications Biology

Reviewer #4 (Remarks to the Author):

The manuscript of Zhang et al. "Spectral Hallmark of Auditory-Tactile Interactions in Somatosensory Cortex" described multisensory interactions between simultaneously presented whisker and auditory stimulation in mice. I was asked by the editor to assess responses of the authors to concerns of the Reviewer #1, as he or she was not currently available to comment.

Major concerns:

In my view, both major concerns raised by the reviewer #1 were adequately addressed. The authors have provided data supporting the invariance of sound-driven inhibition of tactile responsive neurons in S2. The results were included into the manuscript (Supp. Fig 7.).

The second major concern, categorization into 'highpass' and 'bandpass' neurons, were completely removed from the manuscript and replaced by low- and high-frequency tuned neurons as asked by the reviewer #1.

I can good understand both concerns of the previous reviewer. New figures have extensively addressed these concerns and increased conclusiveness of the presented results.

Minor concerns:

The minor concern #1 was an important comment, as in previous version of the manuscript, most probably, no comprehensive description of the method was provided. In the recent version, the methodology of identification of responsive neurons is understandable, although the description could still be improved for better and faster intelligibility.

All other minor concerns were also acceptably addressed.

To conclude, all questions and concerns were satisfactorily responded and addressed in the manuscript. The manuscript expands our understanding of neuronal multimodal integration using a sound methodology, and therefore could be recommended for publication. The study analyzed data collected from the neurons in the cortical layer 2/3, which is pointed out by the authors in the manuscript, but it is not mentioned in discussion. Consideration of cortical layers 1 and 6, which are specifically involved in mechanisms of sensory gating and informational flow modulation, would be also appropriate along with a discussion about possible effects of attention and arousal.

**Responses to Reviewers (COMMSBIO-19-1607-T)
January 10, 2020**

Reviewer #2 (Remarks to the Author):

The authors have satisfactorily addressed my questions, I recommend this paper for publication in Communications Biology

Thank you.

Reviewer #4 (Remarks to the Author):

The manuscript of Zhang at al. "Spectral Hallmark of Auditory-Tactile Interactions in Somatosensory Cortex" described multisensory interactions between simultaneously presented whisker and auditory stimulation in mice. I was asked by the editor to assess responses of the authors to concerns of the Reviewer #1, as he or she was not currently available to comment.

Major concerns:

In my view, both major concerns raised by the reviewer #1 were adequately addressed. The authors have provided data supporting the invariance of sound-driven inhibition of tactile responsive neurons in S2.

The results were included into the manuscript (Supp. Fig 7.).

The second major concern, categorization into 'highpass' and 'bandpass' neurons, were completely removed from the manuscript and replaced by low- and high-frequency tuned neurons as ask by the reviewer #1.

I can good understand both concerns of the previous reviewer. New figures have extensively addressed these concerns and increased conclusiveness of the presented results.

Thank you for stepping in to referee the manuscript, and we are happy to see we were able to address these concerns adequately.

Minor concerns:

The minor concern #1 was an important comment, as in previous version of the manuscript, most probably, no comprehensive description of the method was provided. In the recent version, the methodology of identification of responsive neurons is understandable, although the description could still be improved for better and faster intelligibility.

We tweaked the description in the Methods in a way that should be a bit easier to digest.

All other minor concerns were also acceptably addressed.

To conclude, all questions and concerns were satisfactorily responded and addressed in the manuscript. The manuscript expands our understanding of neuronal multimodal integration using a sound methodology, and therefore could be recommended for publication. The study analyzed data collected from the neurons in the cortical layer 2/3, which is pointed out by the authors in the manuscript, but it is not mentioned in discussion. Consideration of cortical layers 1 and 6, which are specifically involved in mechanisms of sensory gating and informational flow modulation, would be also appropriate along with a discussion about possible effects of attention and arousal.

Thank you for this suggestion. We now mention this point in the discussion and reference studies that looked at crossmodal and neuromodulatory inputs to layers 1 and 6.